# Differential regulation of the *Epr3* receptor coordinates membrane-restricted rhizobial colonization of root nodule primordia

Yasuyuki Kawaharada[1,†], Mette W. Nielsen[1], Simon Kelly[1], Euan K. James[2], Kasper R. Andersen[1], Sheena R. Rasmussen[1], Winnie Füchtbauer[1], Lene H. Madsen[1], Anne B. Heckmann[1,†], Simona Radutoiu[1] & Jens Stougaard[1]

In *Lotus japonicus*, a LysM receptor kinase, EPR3, distinguishes compatible and incompatible rhizobial exopolysaccharides at the epidermis. However, the role of this recognition system in bacterial colonization of the root interior is unknown. Here we show that EPR3 advances the intracellular infection mechanism that mediates infection thread invasion of the root cortex and nodule primordia. At the cellular level, *Epr3* expression delineates progression of infection threads into nodule primordia and cortical infection thread formation is impaired in *epr3* mutants. Genetic dissection of this developmental coordination showed that *Epr3* is integrated into the symbiosis signal transduction pathways. Further analysis showed differential expression of *Epr3* in the epidermis and cortical primordia and identified key transcription factors controlling this tissue specificity. These results suggest that exopolysaccharide recognition is reiterated during the progressing infection and that EPR3 perception of compatible exopolysaccharide promotes an intracellular cortical infection mechanism maintaining bacteria enclosed in plant membranes.

[1] Department of Molecular Biology and Genetics, Centre for Carbohydrate Recognition and Signalling, Aarhus University, Gustav Wieds Vej 10, Aarhus C DK-8000, Denmark. [2] The James Hutton Institute, Invergowrie, Dundee DD2 5DA, UK. † Present addresses: Department of Plant Bio Sciences, Faculty of Agriculture, Iwate University, 3-18-8 Ueda, Morioka, Iwate, Japan (Y.K.); Arla Foods Ingredients, Sønderhøj 10, 8260 Viby J, Denmark (A.B.H.). Correspondence and requests for materials should be addressed to J.S. (email: stougaard@mbg.au.dk).

Endosymbioses with rhizobia enable legumes to fix atmospheric nitrogen and in return rhizobia find a niche for their growth and multiplication within legume nodules. Rhizobial infection of the model legumes *Lotus japonicus* (*Lotus*) and *Medicago truncatula* (*Medicago*) is initiated at sites where bacteria are entrapped by curled root hairs. Recent results suggest that an infection chamber is formed through remodelling of the plant cell wall before infection threads are initiated and colonized by rhizobia[1]. These plant-derived tubular structures traverse inwards to the root cortex where they ramify in nodule primordia, which are formed by associated local re-initiation of cortical root cell divisions. Finally, bacteria are released into nodule cells by endocytosis. By this process, membrane-bound organelle-like symbiosomes containing nitrogen-fixing rhizobia, in a bacteroid differentiated state, will eventually occupy most of the cytoplasmic space of infected nodule cells. This intracellular invasion mechanism contains bacteria within plant membranes from the initial invagination of the root hair membrane to the symbiosomes residing within the plant cells. Intracellular infection is, however, not the only entry mechanism found in the large and diverse legume family. Crack entry between epidermal cells, often associated with lateral root formation, is an alternative intercellular route known from a number of legumes, for example, peanut (*Arachis hypogaea*), *Aeschynomene* and *Sesbania rostrata*[2–4]. In some of these legumes rhizobia remain intercellular until symbiosomes are formed by endocytosis, whereas in others cortical infection threads develop from the initial infection pockets re-establishing an intracellular entry mode in the root cortex[3]. Both of these alternative crack entry pathways were also found in *Lotus* mutants where the normal root hair infection thread pathway was blocked, although infection efficiencies were low compared to intracellular infection via root hair infection threads[5–7].

Genetic analysis of loss-of-function and gain-of-function plant mutants supports the existence of two parallel, but highly synchronized, pathways controlling intracellular root hair infection and nodule organogenesis[7,8]. However, the plant genetic program controlling intracellular infection via infection threads remains uncharacterised. In *Lotus*, both root hair infection and nodule organogenesis pathways are under the control of the LysM-domain-containing Nod factor receptors, NFR1 and NFR5, that trigger two signal transduction pathways following the perception of rhizobial Nod factor signals[9–13]. The so-called common symbiosis pathway, shared between the rhizobial and mycorrhizal symbioses, consists of at least seven genes[14,15]; a LRR receptor kinase (*SymRK*)[16,17], cation channels (*Castor* and *Pollux*)[18–20], nucleoporins (*Nup85*, *Nup133* and *Nena*)[21–23] and a calcium calmodulin-dependent protein kinase (*CCaMK*)[24,25]. *SymRK*, *Castor*, *Pollux*, *Nup85*, *Nup133* and *Nena* all act upstream of $Ca^{2+}$ spiking. Biochemical analysis suggests that NFR receptors bind Nod factor and form a complex where NFR5 interacts with SYMRK[26–28]. The current understanding is that this receptor activation leads to the release of calcium oscillations in the nucleoplasm[29,30]. Calcium oscillations are decoded by CCaMK and activated CCaMK phosphorylates CYCLOPS that acts as a transcription factor for *Nin*, thus promoting epidermal infection thread formation[31,32]. Further downstream in the organogenesis pathway the NSP1, NSP2, NIN, NF-YA1 and the AP2 type ERN1 and ERN2 transcriptional regulators are required for regulation of nodule-expressed genes and initiation of nodulation[33–41]. Finally, reactivation of cortical root cells resulting in nodule primordia formation is initiated by cytokinin signalling through functionally overlapping cytokinin receptors LHK1, LHK1a and LHK3 (refs 42–44). Endoreduplication involving topoisomerase (*Vag1* and *SUNERGOS*) and a deubiquitination enzyme (*Amsh1*) is also required[45–47].

A comparable progression of events cannot be assembled from the few components identified to contribute to infection thread development and progression. *Npl1* encodes a pectate lyase presumably involved in cell wall hydrolysis[48], *Nap1*, *Pir1*, *ArpC1* and *SCARN* encode proteins controlling actin rearrangement in root hairs via the SCAR/WAVE complex[49–52], and a putative E3 ubiquitin ligase encoded by *Cerberus* is also required for infection thread development[53,54]. Infection threads are typically arrested in the root hair if these genes are mutated. Likewise, the transcriptional regulators CYCLOPS, NIN, NSP1 and NSP2 are required for the epidermal initiation of infection thread development. The biochemical pathway and genetic network subsequently involved in progressing infection threads into the root cortex is virtually undescribed and our understanding is mainly based on imaging. Individual plant cells initiate infection thread formation at the interface of the cell above and extend the infection thread to the interface of the cell below. Analysis of electron micrographs suggests that fusion of the infection thread wall at the site of entry into the lower cell precedes cell wall degradation and re-initiation of the infection thread in the lower cell[55]. This iterated cell autonomous process, which appears to differ from the initial infection chamber formation, advances the infection thread and, by an unknown mechanism, branching occurs in the nodule primordium before bacteria are released from the infection thread into the plant cell. Infection thread progression is synchronized and coupled to the development of primordia in a highly regulated process that has not yet been uncovered. An example of this is the abortion of most infection threads already in the epidermis. This leads to the notion of an epidermal–cortical barrier where cytokinin signalling mediates repression[56], while endoreduplication promotes reinitiation[45,46], suggesting that infection thread progression is regulated at each cell passage. Calcium oscillations observed in the plant cell just ahead of the growing infection thread suggest that Nod factor perception and CCaMK activation is involved but components of a regulatory mechanism controlling the cell autonomous advance of infection threads have not yet been identified.

Here we show that perception of exopolysaccharide (EPS) synthesized by *M. loti* is important for maintaining an intracellular infection mode and that expression of the EPS receptor gene, *Epr3*, is controlled by the symbiotic program of the plant. EPR3 acts in the root cortex and nodule primordia to support and sustain the containment of rhizobia and to facilitate an efficient infection process.

## Results

**EPS and EPR3 determine the cortical infection mechanism.** Perception of EPSs by the EPR3 receptor of wild-type *Lotus* (ecotype Gifu) regulates epidermal infection. *epr3* mutants formed fewer root hair infection threads after inoculation with *M. loti* strain R7A than Gifu. Similarly, reduced root hair infection thread formation was observed after Gifu was inoculated with an EPS-deficient R7AexoB mutant[57]. To assess the role of EPS perception in later stages of the nodule infection process, we traced the consequences of perturbed EPS perception on rhizobial invasion of the root cortex and nodule primordia. The infection phenotypes of cortical nodule primordia and nodules were investigated in detail using light microscopy, confocal microscopy and transmission electron microscopy (TEM). When sections of infected nodules of R7A-inoculated *epr3* plants were investigated, misguided infection threads associated with balloon-like swellings were often observed in epidermal cells, some of which reinitiated infection thread elongation into the primordia (Fig. 1a–d). Infection threads in Gifu plants inoculated with R7AexoB appeared thicker and seemed arrested or temporarily arrested at

the cortical boundary (Fig. 1e,f). These observations suggest a control point for infection thread progression at the epidermal–cortical cell boundary and that EPS perception by EPR3 might be required for reinitiating infection threads at each cell passage.

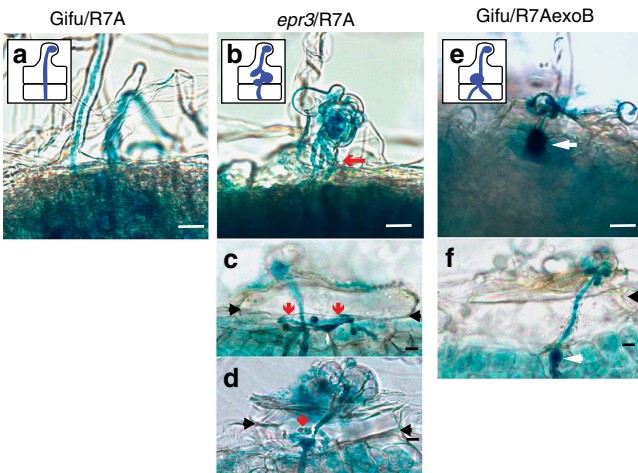

**Figure 1 | Infection thread progression in Gifu and *epr3* mutants.**
(**a**) Schematic and image of infection thread targeting
a nodule primordium in Gifu inoculated with R7A. (**b–d**) Schematic and images of misguided infection threads in *epr3-9* (**b**), *epr3-10* (**c**) and *epr3-11* (**d**) inoculated with R7A. (**e,f**) Schematic and images of short thick infection threads in Gifu inoculated with R7AexoB. (**a–f**) The behaviour of infection threads at the epidermal–cortical cell boundary in Gifu and *epr3* mutants inoculated with R7A or R7AexoB. Red arrows indicate misguided infection threads at the epidermal–cortical boundary (which is marked by black arrows). White arrows indicate balloon-like swellings. Scale bars, 20 µm (**a–f**).

Corroborating these observations, we found that the highly efficient intracellular infection through infection thread progression was impaired in the *epr3* mutants and intercellular bacteria were present in most nodules (Fig. 2; Supplementary Fig. 1). At 14 and 28 days post inoculation (dpi), *epr3* mutant nodules contained fewer infected cells than Gifu. Direct quantification found a significant reduction of transcellular cortical infection threads induced by R7A in *epr3* nodules compared to Gifu nodules (Fig. 2o). Furthermore, transcellular infection threads often appeared irregular with lumpy protrusions in *epr3* mutants compared with Gifu (Fig. 2g,i,k,m; Fig. 3a–d,i,j). The TEM images show enlarged infection threads packed with rhizobia both between cells and within plant cells as well as intercellular bacteria (Supplementary Fig. 1l). In a complementary experiment where Gifu was inoculated with R7AexoB, which lacks EPS, comparable effects were observed. Intercellular infection and a reduction in cortical infection threads were characteristic for Gifu nodules induced and infected by R7AexoB (Fig. 2j,n,o). More detailed microscopy, focusing on infection threads and endocytosis into plant cells, revealed a release mechanism similar to the single-cell peg infections previously reported[7] (Fig. 2n; Fig. 3m–p). Both the absence of EPS in R7AexoB (Fig. 3m–p) and inactivation of *Epr3* (Fig. 3e–l) appear to result in plant cell invasion via this entry mode. Typically, the effects of the absence of EPS or the EPR3 receptor are fewer infected nodule cells, reduced or perturbed transcellular cortical infection threads, and rhizobia present in the intercellular space between host cells (Fig. 2i,j,m,n; Supplementary Fig. 1l). These changes were even more pronounced in plants grown at 28 °C with several bacteroids enclosed within the same symbiosome membrane, which in many cases appeared to be undergoing degradation (Supplementary Fig. 1). This concurs with earlier observations suggesting a role for EPS perception at the initiation of infection threads and bacterial release from infection threads[57,58].

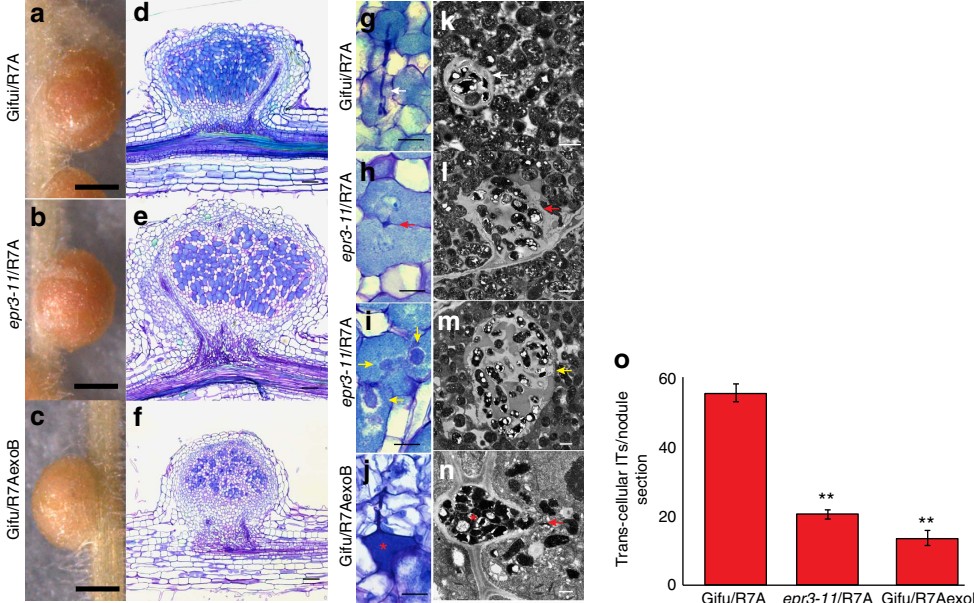

**Figure 2 | Nodule infection phenotype in Gifu and *epr3* mutants.** Images of intact nodules, semi-thin microscopy sections and TEM sections from Gifu inoculated with R7A (**a,d,g,k**), *epr3-11* inoculated with R7A (**b,e,h,i,l,m**) and Gifu inoculated with R7AexoB (**c,f,j,n**). Plants were grown at 21 °C and nodules sectioned 14 dpi. Light microscopy and transmission electron micrographs of sectioned nodules illustrate the infection mechanisms (**g–n**). White arrows indicate transcellular infection threads. Red arrows indicate peg-type intercellular infection. Yellow arrows indicate infection thread protrusions. Red stars indicate the intercellular bacteria. Scale bars, 0.5 mm (**a–c**); 0.1 mm (**d–f**); 20 µm (**g–j**); 1 µm (**k–n**). (**o**) Number of cortical infection threads observed in nodules at 14 days post infection (dpi). Gifu and *epr3-11* inoculated with R7A and Gifu inoculated with R7AexoB. Number of cortical infection threads was counted in randomly selected sections of five nodules from each combination. ** ($P < 0.01$, Student's *t*-test, $n = 25$) indicates significant difference compared to Gifu inoculated with R7A. Error bars are s.e.m.

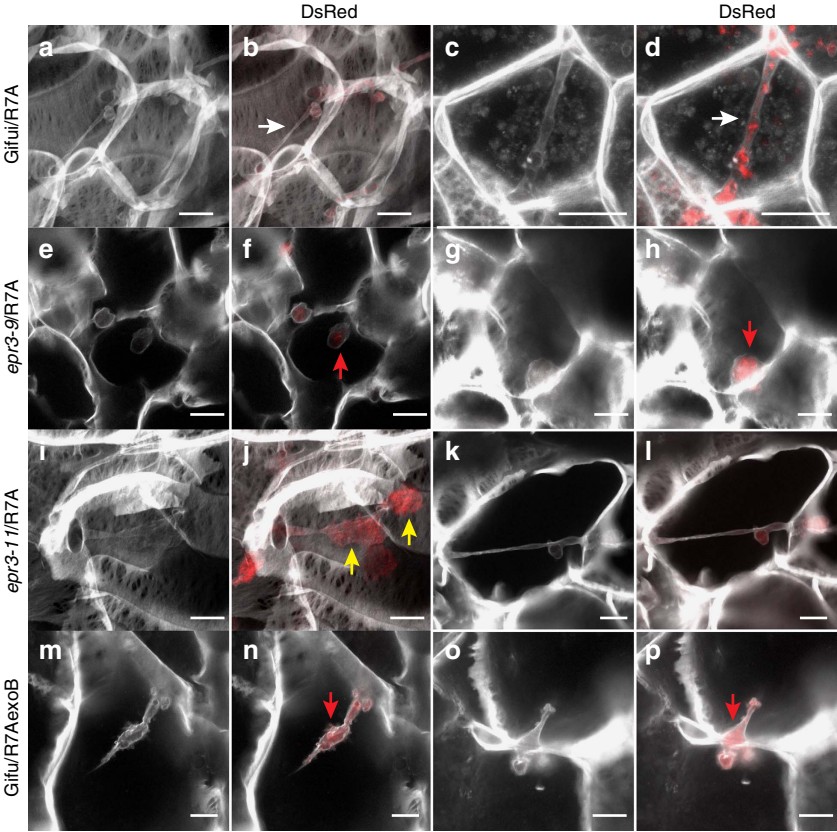

**Figure 3 | Cortical infection threads and peg infection in Gifu and *epr3* mutants.** Images of cortical infection threads in nodules of Gifu inoculated with R7A DsRed (**a**–**d**), *epr3-9* mutant inoculated with R7A DsRed (**e**–**h**), *epr3-11* inoculated with R7A DsRed (**i**-**l**) and Gifu inoculated with R7AexoB DsRed (**m**-**p**). At 10–14 dpi, nodules were sectioned by hand and cell walls, and infection threads were stained with 0.04% calcofluor. White arrows indicate transcellular infection threads. Red arrows indicate peg-type intercellular infection. Yellow arrows indicate infection thread protrusions. Scale bars, 10 µm.

Taken together, the impaired infection of *epr3* mutant nodules by R7A and Gifu nodules induced by R7AexoB indicate that EPS-mediated EPR3 signalling is required for sustaining the cortical infection thread invasion that normally mediates intracellular rhizobial infection from epidermal passage to endocytotic release into symbiosomes.

**The cellular *Epr3* expression pattern mirrors infection.** Previous observations showed *Epr3* promoter activation in root hairs of the susceptible zone after Nod factor perception[57]. To determine whether the suggested cortical function is reflected in the *Epr3* expression pattern, we monitored *Epr3* promoter activity using nuclear localized fluorescence in transformed roots expressing the *pEpr3:tYFP-NLS* construct. A representative root is shown in Fig. 4 covering a time span from 3 to 10 dpi after inoculation with *M. loti* MAFF303099. At 3 dpi, the expression was fairly uniform among root hairs. By 4 dpi, an expression focal point(s) was distinguishable above what appears to be an emerging nodule primordium. From 5 to 10 dpi, the expression in root hairs declined, while increased expression in the developing primordia abutted an area where DsRed-labelled rhizobia were observed. A similar pattern was observed using a *pEpr3:GUS* reporter construct (Supplementary Fig. 2a–f). At 7 and 14 dpi, the expression was confined to primordia and nodules, indicating organ specialization. The *Epr3* promoter activity was lower and ceased in older nodules (Supplementary Fig. 2e). In whole mounts/sections *pEpr3:tYFP-NLS* expression was strong in infected epidermal cells (Fig. 5a,b) and in outer cortical cells immediately surrounding infection threads (Fig. 5b).

This high level of expression appears to delineate the infection thread path from the epidermis into the outer root cortex and the nodule primordia.

Previous results showed that *Epr3* expression was rapidly induced in the epidermal cells after Nod factor application or rhizobia inoculation[57]. Considering the developmental expression pattern of *Epr3* described above, we asked whether *Nfr1*, *Nfr5* and *Epr3* would be co-expressed and hence whether *Nfr1* and *Nfr5* could be involved in the cellular regulation in both root epidermis and cortex. First, *pEpr3:GUS* was transformed into *nfr1* and *nfr5* mutants to assay *Epr3* promoter activation. The absence of reporter gene activity in these genotypes shows that *Epr3* promoter activation is dependent on both *Nfr1* and *Nfr5* (Fig. 6c; Supplementary Table 1). Second, the expression of *Nfr1* and *Nfr5* was monitored using promoter GUS reporters. Both promoters were active in uninoculated root epidermal cells, as expected for Nod factor receptors (Supplementary Fig. 2g,k). Additional detailed studies using *pNfr1:tYFP-NLS* and *pNfr5:tYFP-NLS* constructs confirmed whole root expression in the uninoculated roots (Supplementary Fig. 3). The *Nfr1* promoter was generally found to have a lower level of activity compared to *Nfr5*. In inoculated roots, the *Nfr1* and *Nfr5* promoters remained active within infected root hairs, but activity of the *Nfr1* promoter was difficult to detect using the *tYFP-NLS* and *GUS* fusion constructs due to the low expression level (Fig. 5e; Supplementary Fig. 2h,l). At later time points, this epidermal expression weakened and was followed by expression in the cortex as the development of nodule primordia progressed. No *Epr3*, *Nfr1* or *Nfr5* promoter activity was observed in fully developed nodules (Fig. 5d,h,l; Supplementary Fig. 2e,j,m).

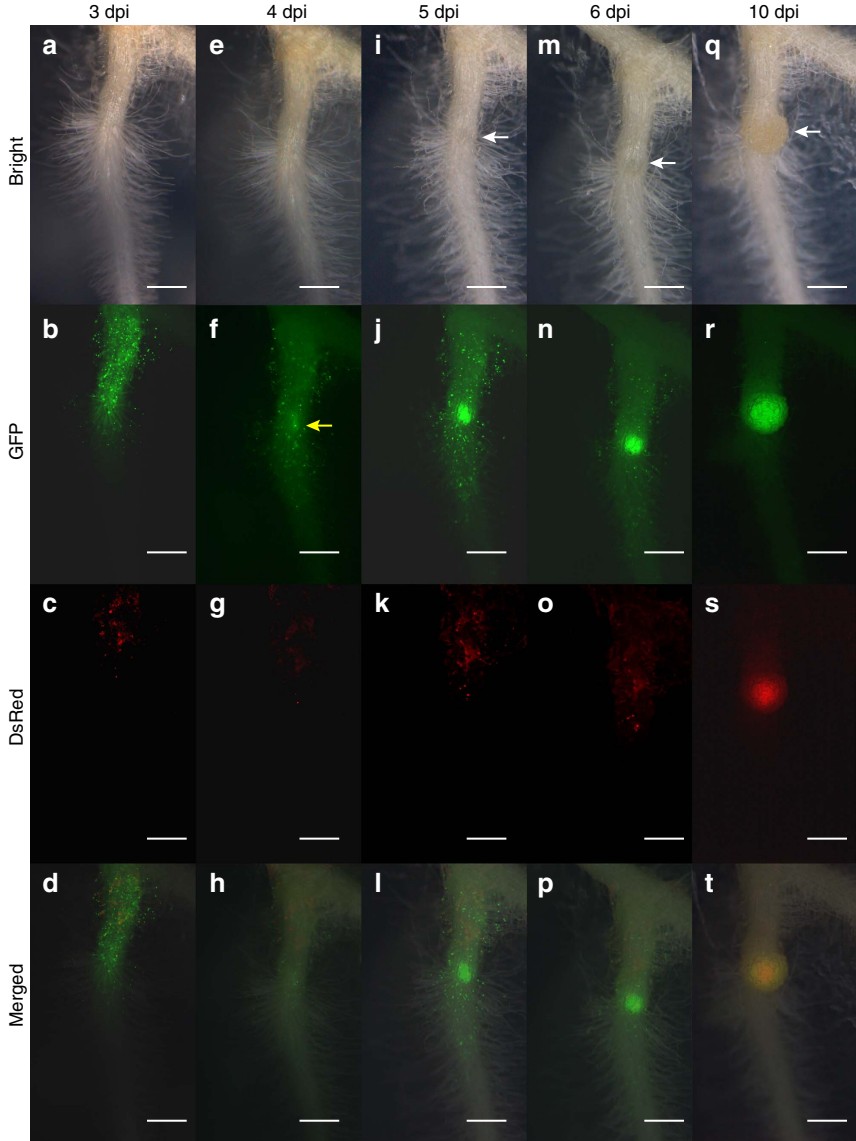

**Figure 4 | Temporal and spatial expression of *pEpr3:tYFP-NLS* activity in Gifu.** Time series on the same root from 3 to 10 dpi with *M. loti* MAFF303099 DsRed. **a–d** are 3 dpi, (**e–h**) 4 dpi, (**i–l**) 5 dpi, (**m–p**) 6 dpi and (**q–t**) 10 dpi. (**a,e,i,m,q**) Bright-field images, (**b,f,j,n,r**) GFP filter images and (**c,g,k,o,s**) DsRed filter images. (**d,h,l,p,t**) Merged bright, GFP and DsRed images. White arrows indicate nodule development and yellow arrow indicate strong *pEpr3:tYFP-NLS* expression in the epidermal cell in response to *M. loti*. Scale bars, 0.5 mm.

These results show that both *Nfr1* and *Nfr5* are expressed in the cells, where *Epr3* is induced in the root cortex. This suggests that the two-step recognition signalling mediated by Nod factor and EPS perception by LysM receptor proteins at the epidermal root interface may also be operative in the cortical cell layers and in nodule primordia as the rhizobial infection progresses.

**Symbiotic transcription factors coordinate *Epr3* expression.** Studies in *Lotus* and *Medicago* have contributed to the understanding of a symbiotic signal transduction pathway acting downstream of the Nod factor receptors[59]. Activation of this pathway is critical for initiating infection and/or organogenesis[7]. We took advantage of available loss-of-function mutants inactivating these pathways and the *pEpr3:GUS* reporter to examine whether *Epr3* was independently regulated or integrated into these pathways (Fig. 6). Initially, the role of genes primarily required for infection thread formation and/or progression was

investigated in *cerberus*, *nap1* and *pir1* mutants transformed with *pEpr3:GUS*. Reporter gene activity was detected at levels and locations similar to Gifu, indicating that these genes were not essential for *Epr3* expression (Supplementary Table 1). However, reporter gene expression driven by the *Epr3* promoter was absent in mutants acting upstream of $Ca^{2+}$ spiking (*nfr1*, *nfr5*, *symrk*, *nup133*, *pollux* and *nena*) and in *ccamk* mutants impaired in their ability to interpret the calcium oscillations (Fig. 6; Supplementary Table 1). Nod factor perception and the onset and interpretation of $Ca^{2+}$ spiking through the common symbiosis pathway were thus required for *Epr3* activation. Similar analyses in mutants lacking symbiotic transcriptional activators revealed a differential genetic requirement for the epidermal and cortical onset of *Epr3* transcription. We found no detectable epidermal expression in *nsp2* and *ern1* mutants, a very reduced expression in *cyclops*, while *Nsp1* and *Nin* were dispensable for epidermal expression (Fig. 6). Further evidence for differential regulation of *Epr3* expression in the epidermal and cortical primordia was obtained

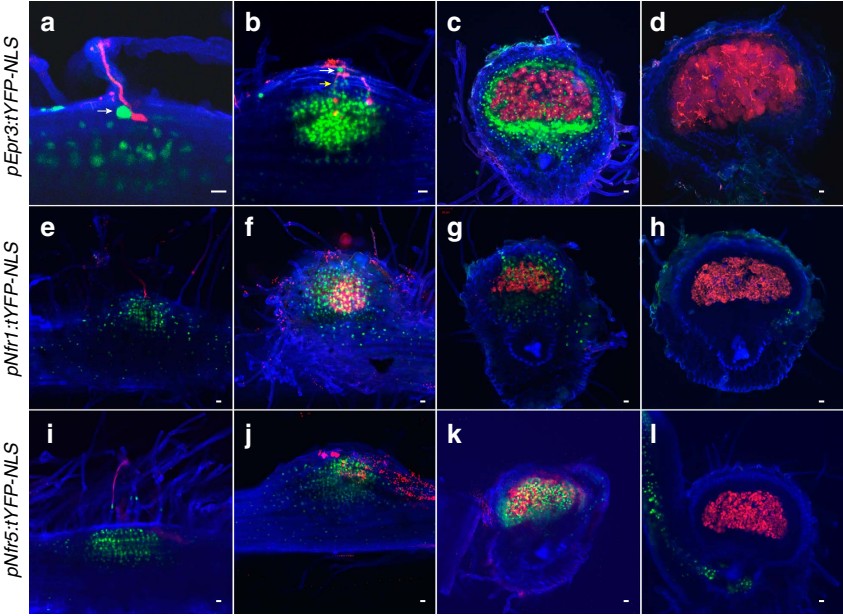

**Figure 5 | Temporal and spatial expression of *Epr3* and *Nfr* promoter reporter constructs.** Images of the infection process from the root hair infection stage (**a,e,i**) to nodule maturity (**d,h,l**) were collected from sections/whole mounts of transgenic roots inoculated with *M. loti* MAFF303099 DsRed. (**a–d**) *pEpr3:tYFP-NLS*, (**e–h**) *pNfr1:tYFP-NLS* and (**i–l**) *pNfr5:tYFP-NLS*. White and yellow arrows show specific *Epr3* expression in epidermal and outer cortical cells adjacent to infection threads. Scale bars, 20 μm.

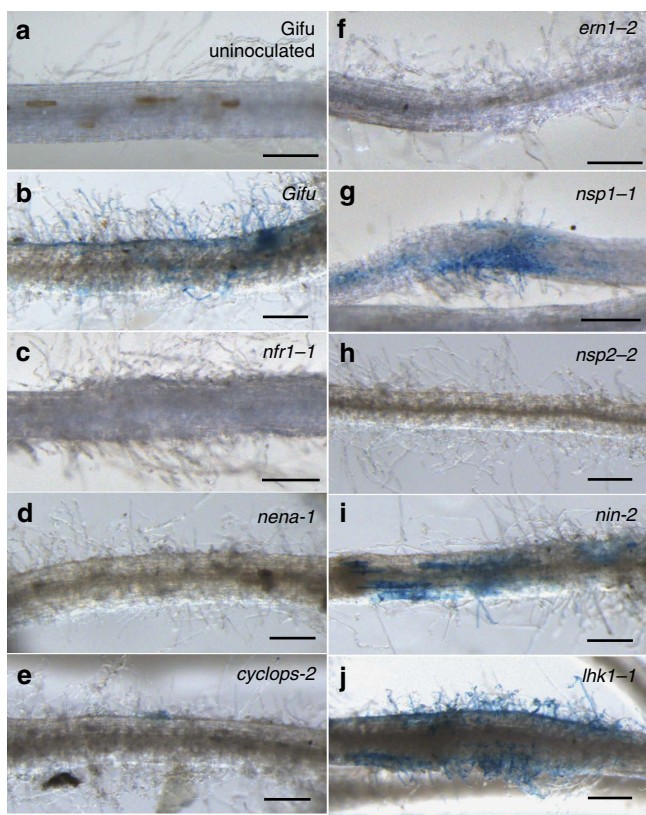

**Figure 6 | *Epr3* promoter activation and transcript levels in symbiotic mutants.** Images of transgenic roots stained for *pEpr3:GUS* activation in Gifu and symbiotic *Lotus* mutants inoculated with R7A and assayed 3–10 dpi. (**a**) Uninoculated Gifu control (**b**), inoculated Gifu, (**c**) *nfr1-1*, (**d**) *nena-1*, (**e**) *cyclops-2*, (**f**) *enr1-2*, (**g**) *nsp1-1*, (**h**) *nsp2-2* (**i**) *nin-2* and (**j**) *lhk1-1*. Scale bars, 250 μm. See Supplementary Table 1 for the complete data set and the number of plants analysed.

from *nena*, *cyclops*, *nsp1* and *ern1* mutants, which eventually initiate the organogenesis program in the cortical layers leading to nodule primordia formation, and from the *snf1* mutant *(CCaMK* gain of function; Supplementary Fig. 4; Supplementary Table 1). In these mutants, the *Epr3* promoter was active in the emerging nodule primordia. We then investigated whether promoter sequences mediating epidermal and cortical *Epr3* expression could be delimited in transgenic Gifu roots using a set of promoter deletions. In these Gifu roots, the 1,019 (ref. 57), 684, 329 and 280 bp promoters induced GUS expression in both the epidermis and nodule primordia. However, further deletion of promoter sequences to 257 bp eliminated epidermal expression while activation in nodule primordia was still observed (Fig. 7a,c). This analysis positions a regulatory promoter element sufficient for epidermal expression between 280 and 257 bp, and an element regulating primordial expression within the 257 bp promoter. Mutation of a sequence in the 280 bp promoter segment (mutation 1) corresponding to a previously characterized AP2 transcription factor binding site[60] eliminates epidermal *Epr3* expression (Fig. 7b,c). In contrast, no detectable change in epidermal expression was observed after mutation of a sequence corresponding to a previously identified NIN-binding site[39,61] in the 280 promoter (mutation 2). However, mutation of this NIN1-binding site in the 257 bp promoter eliminates the cortical expression (Fig. 7b,c), suggesting that ERN1 can still activate the *Epr3* 280 promoter lacking the NIN1-binding site (mutation 2) in the cortex. The role of the predicted ERN1- and NIN1-binding sites of the 280 bp *Epr3* promoter segment (Fig. 7b) was further investigated in an electrophoretic mobility shift assay (EMSA; Fig. 7f). Full-length ERN1 and truncated NIN (residues 520–878) proteins were expressed in *Escherichia coli* and binding of DNA probes encompassing the corresponding binding sites was determined as mobility shifts (Fig. 7b,f; Supplementary Table 2). Mobility shifts were observed for both the ERN1 and NIN proteins, and binding of DNA fragments carrying mutations found to impair epidermal or cortical expression in *pEpr3:GUS* analyses was eliminated or reduced (Fig. 7f).

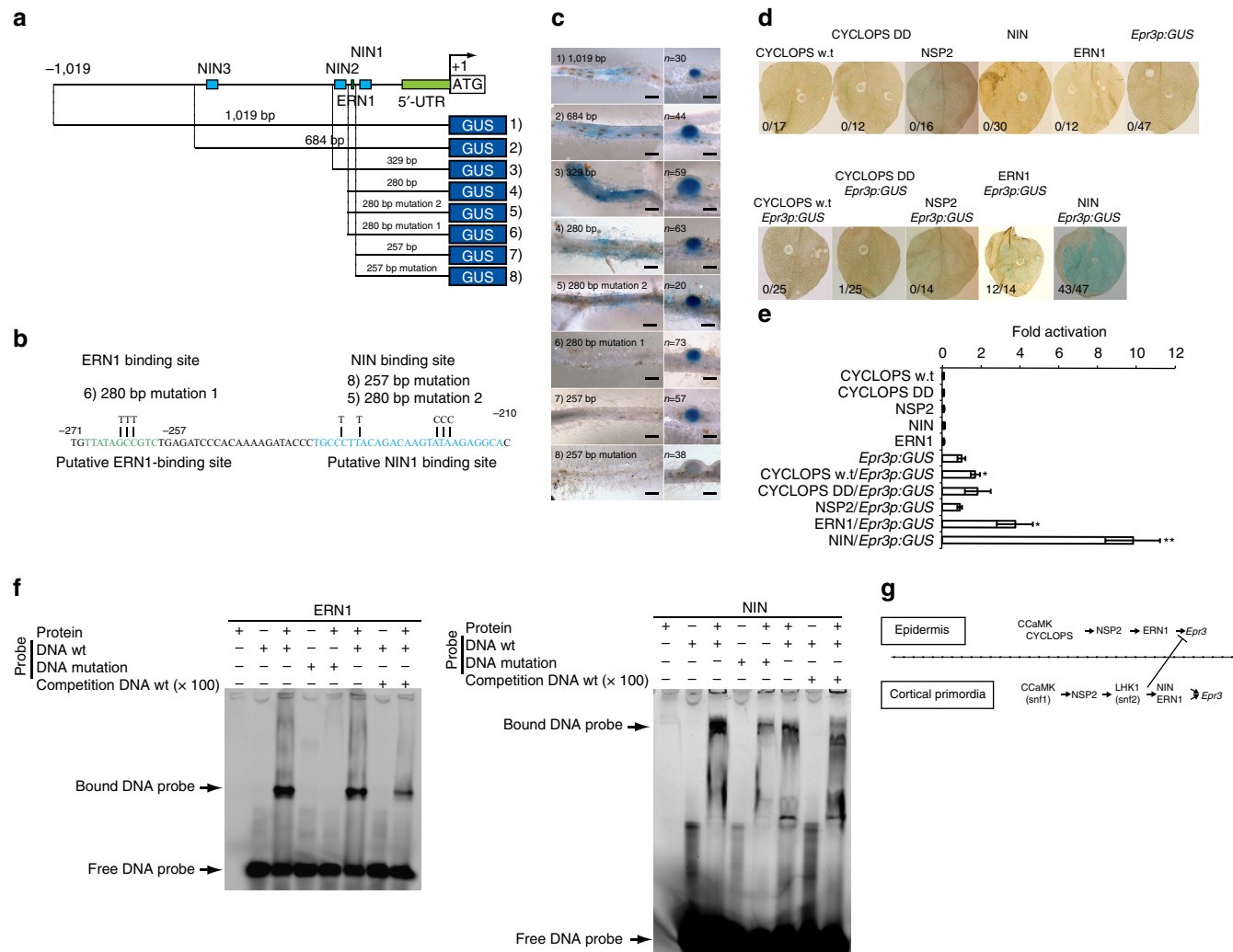

**Figure 7 | *Epr3* promoter analysis in Gifu.** (**a**) Overview of the *Epr3* promoter region and the analysed deletion series. Positions of three putative NIN-binding sites and one putative ERN1-binding site are indicated. (**b**) Nucleotide changes in the mutated putative binding sites for ERN1 and NIN. (**c**) Gifu plants transformed with *pEpr3:GUS* promoter deletions were inoculated with R7A and assayed at 4–10 dpi. Panels show *Epr3* expression in epidermal and cortical cells. Scale bars, 0.5 mm. (**d**) The *pEpr3:GUS*, and/or constructs for overexpression of *Cyclops*, *Cyclops DD*, *Nsp2*, *Ern1* and *Nin* were infiltrated into *N. benthamiana* leaves. Upper row shows single infiltration, and bottom row is co-infiltration of *pEpr3:GUS* together with CYCLOPS, CYCLOPS DD, NSP2 or NIN. After 3 days incubation, these leaves were stained by X-gluc, and destained. Numbers indicate fraction of GUS-positive leaves. (**e**) Levels of GUS activity induced by *pEpr3:GUS* in *N. benthamiana* leaves after 3 days infiltration. Panel shows fold change compared to *pEpr3:GUS* single infiltration. Error bars are s.e. * or ** ($P < 0.05$ or $P < 0.01$, Student's *t*-test, $n = 4–5$) indicates significant difference compared to *pEpr3:GUS* single infilutaration. (**f**) EMSAs were performed using His-tagged ERN1 and truncated NIN (residues 520–878) together with putative ERN1- and NIN-binding DNA sequences. Sequences of DNA wt and mutant versions of the *Epr3* promoter corresponds to **b**, and DNA probes were FAM-tagged. Competition DNA is 100 times concentration of wt DNA without FAM. (**g**) Model of *Epr3* induction. In the epidermis, ERN1 is sufficient for *Epr3* expression downstream of CCaMK, CYCLOPS and NSP2. In the cortical primordia, both NIN and ERN1 induce *Epr3* expression directly downstream of CCaMK, NSP2 and the LHK1 cytokinin receptor. Cytokinin signaling through LHK1 has a negative effect on epidermal *Epr3* expression possibly involving the mechanism regulating infection thread formation[43].

Elevated cytokinin levels and signalling have been shown to regulate the number of infection threads progressing into the root cortex[43,62]. We asked whether *Epr3* expression would be regulated by cytokinin signalling. Gifu roots were treated with $10^{-8}$ M BAP and the *Epr3* transcript was quantified after 3, 6 or 12 h. In contrast to Nod factor treatment, application of cytokinin did not induce detectable *Epr3* expression in roots during this short-term treatment (Supplementary Fig. 5a), while *Epr3* was expressed in spontaneous nodule primordia formed by *snf2* (Supplementary Fig. 4h,i). Additional insight into the role of cytokinin signalling on *Epr3* expression was obtained from the analysis in the loss-of-function cytokinin receptor *lhk1-1* mutant. In the *lhk1-1* genetic background, the *Epr3*:GUS activity was abundant in the root hairs/epidermis compared to wild type, at the same time point after inoculation (Fig. 6j). Likewise, the overall *Epr3* transcript level was higher in *lhk1-1* mutant roots at 3 and 14 dpi compared to wild type (Supplementary Fig. 5b,c). This indicates that cytokinin signalling is, directly or indirectly, involved in repression of epidermal *Epr3* expression and correlates the *Epr3* expression level to the frequency of infection thread progression known to be increased in *lhk1-1* mutants[43].

Our detailed analyses of the onset of *Epr3* transcription in various symbiotic mutants revealed that the activation of the EPS recognition step has a differential genetic requirement in the epidermal cell layer, compared to the root cortex where nodule primordia emerge. The increased *Epr3* expression in *lhk1-1*

mutants and induction of *Epr3* in primordia of *snf2* mutants suggests that cytokinin signalling has opposing effects on *Epr3* expression in epidermal and cortical cells.

**Promoter elements mediate cell layer-specific expression**. Analysis of genetic requirements for *Epr3* promoter activation during root nodule symbiosis revealed that CYCLOPS, NSP2 and ERN1 transcriptional regulators control *Epr3* expression in epidermal root cells. To assess whether this control is exerted through *Epr3* promoter elements, *Epr3* promoter activation was investigated by transient co-expression in *Nicotiana benthamiana* leaves. Constructs expressing CYCLOPS or the autoactive DD version of CYCLOPS[31] previously shown to activate *Nin* expression were infiltrated into *N. benthamiana* leaves together with *pEpr3:GUS*. *Epr3* promoter activation was monitored by staining leaves for GUS activity (Fig. 7d,e). GUS activity was comparable to that of *pEpr3:GUS* alone after co-infiltration of CYCLOPS or the autoactive DD version. Similarly, co-expression of an NSP2 construct and *pEpr3:GUS* did not result in detectable *Epr3* promoter activation. In contrast, co-infiltration of ERN1 and *pEpr3:GUS* resulted in expression of the *Epr3* promoter (Fig. 7d,e). A previous report indicated that *Epr3* is downstream of *Nin*[39,63] while our promoter studies suggested this may only be the case for cortical *Epr3* expression. However, lack of primordia in *nin* mutants prevented detection of *pEpr3:GUS*-based assessment of NIN involvement in cortical expression of *Epr3*. A construct expressing NIN was therefore infiltrated into *N. benthamiana* leaves together with *pEpr3:GUS* and was able to activate the *Epr3* promoter (Fig. 7d,e). These results suggest that ERN1 is essential for *Epr3* expression in both epidermal and cortical cells and delimits an ERN1-binding site mediating epidermal expression of *Epr3* in the epidermis. NIN is essential for induction of *Epr3* expression in cortical cells (Fig. 7g) and a NIN1-binding site sufficient for cortical cell expression was delimited in the *Epr3* promoter. The deletion study did not reveal a major contribution from the putative NIN2- and NIN3-binding sites in the *Epr3* promoter (Fig. 7a). Taken together, our results suggest that ERN1 is sufficient for epidermal *Epr3* expression while NIN appears to be dependent on ERN1 for any epidermal regulation of *Epr3* that may not have been uncovered by our promoter analysis.

**EPS perception leads to efficient symbiotic nodule formation**. The experiments monitoring the capacity of R7A to infect nodules of *epr3* mutants revealed significant qualitative differences to wild-type Gifu. This prompted us to ask whether there were associated quantitative changes in nodulation efficiency. Nodule formation kinetics in Gifu and two allelic *epr3* mutants, *epr3-9* and *epr3-11*, were determined after inoculation with R7A under standard conditions at 21 °C and the infection efficiency estimated from the number/ratio of pink versus white nodules (Supplementary Fig. 6a). A significant delay in infection efficiency resulting in mature nodules was detected in *epr3-9* and *epr3-11* compared to Gifu (Supplementary Fig. 6a). At 10 and 14 dpi, a significant reduction in infected pink nodules was observed on *epr3-9* and *epr3-11* plants compared with Gifu plants, while at 28 dpi, the number of pink nodules on *epr3-9, epr3-11* and Gifu was comparable. To test whether this phenotypic change in infection efficiency was robust, the same experiment was performed at 28 °C. Nodulation of Gifu inoculated with R7A at 28 °C was reduced and delayed compared to 21 °C. However, the development of pink nodules on *epr3-9* and *epr3-11* was further delayed and reduced (Supplementary Fig. 6a). Similar results were obtained with MAFF303099, a highly infective *M. loti* strain (Supplementary Fig. 6b).

Previously, a delay in the nodulation of Gifu at 21 °C was observed with the R7AexoB mutant strain that is unable to synthesize EPS[57,58] and this delay is exacerbated at 28 °C (Supplementary Fig. 6c)[57]. In contrast to R7A-inoculated Gifu plants, no pink nodules had developed by 28 dpi on R7AexoB-inoculated Gifu plants. Furthermore, R7AexoB-inoculated Gifu plants carried significantly fewer pink nodules and more white nodules than R7A-inoculated plants at 35 and 42 dpi. These results indicate that EPS and EPS perception by EPR3 are important for the timely progression of nodule infection and affect nodule development.

**Discussion**
We have shown that the EPR3 LysM receptor kinase perception of rhizobial EPS promotes proficient intracellular infection of root cortical cells and efficient symbiotic interaction between *Lotus* and *M. loti*. In contrast to the *nfr1* and *nfr5* Nod factor receptor mutants, which are strictly required for the initiation of nodulation[9,11,64], *epr3* mutants are nodulated and infected by *M. loti*, albeit with reduced efficiency and with a delay. The mutation of *Epr3* or of EPS synthesis genes in the microbial partner reduces or eliminates nodule infection via infection threads and reduces the number of infected cells. Although a passive physicochemical contribution from EPS cannot be excluded, the phenotypic differences between *epr3* mutants and wild-type plants inoculated with R7AexoB (Supplementary Fig. 6) may suggest that EPR3 has a co-receptor that is partially functional in the absence of EPR3. Microscopy of root nodules shows that *Epr3* is not only necessary for efficient intracellular infection across the epidermis but also in the root cortex and in nodule primordia. Inactivation of *Epr3* results in a reduction of the normal intracellular infection thread mode and increased intercellular infection.

Infection threads releasing bacteria between the primordial cells and crack entry both appear to contribute to the intercellular infection mode. One possibility is that in addition to promoting infection thread formation, EPR3 signalling plays a negative role in the control of crack entry and intercellular infection. In this scenario, a default intercellular infection pathway would take control when *Epr3* is inactivated. Such a regulatory role is supported by the different phenotype of *epr3* mutants compared to other infection thread mutants. In most of these developmental mutants, infection threads are arrested in the root hairs or misguided at the cortical boundary and little or no infection of primordia occurs[21–23,32,45,46,48,50–53].

The *Nfr1* and *Nfr5* Nod factor receptor genes are expressed in epidermal cells of uninoculated roots indicating the necessity for rapid and first-line perception of rhizobial signals by the legume root (Supplementary Fig. 2g,k; Supplementary Fig. 3)[11]. On the other hand, *Epr3* expression is Nod factor-induced (Supplementary Fig. 5a)[57], and *Nfr1*- and *Nfr5*-dependent (Fig. 6c; Supplementary Table 1), positioning EPS-induced EPR3 signalling in a subsequent epidermal recognition step of a two-step surveillance mechanism. We show here that a spatially restricted increase in the expression of the Nod factor and EPS receptors occurs following the advancing symbiotic infection process. The *Nfr1*, *Nfr5* and *Epr3* promoters are activated in root cortical zones where nodule primordia develop (Fig. 5; Supplementary Fig. 2d,i,n), a pattern also observed for *SymRK* and *Lhk1* expression[42,65], indicating receptor controlled signalling throughout rhizobia infection and primordia development. Taking into account that rhizobial *nod* genes are expressed in infection threads and that Nod factor synthesis is likely continuing during infection, this suggests that the two-step Nod factor and EPS recognition process assessing the compatibility of

bacteria accompanies the progression of the infection thread from the epidermis into the cortical roots cells and nodule primordia. This interpretation is supported by the results obtained using a promoter with enriched epidermal activity to express *Nfr1* in *Lotus* and *Nfp* (the *Nfr5* homologue) in *Medicago* in their corresponding mutant backgrounds. In both experiments, cortical cell division was induced, but infection thread formation was restricted[8,66]. Furthermore, calcium spiking in the root cortex has been shown to follow the progressing infection thread[30].

Symbiotic mutant analyses in *Lotus* have allowed for the distinction of genes involved in the two pathways controlled by NFR1 and NFR5 following Nod factor perception, root hair infection and cortical organogenesis[7]. Our analysis reveals that *Epr3* promoter activation takes place independently in both branches of the Nod factor signalling pathway and is dependent on genes acting before calcium spiking. *Cyclops*, *Nsp2* and *Ern1*, which control the root hair infection pathway[31–34,36,40,41,67], also control early *Epr3* activation (Fig. 6), but not the later onset of transcription in *cyclops* and *ern1* nodule primordia (Supplementary Fig. 4). *Epr3:GUS* activation after co-expression in *N. benthamiana* and EMSAs suggest that ERN1 regulates *Epr3* directly, while CYCLOPS and NSP2 act indirectly. The *Epr3* promoter analyses suggest that *Epr3* expression in the epidermis is regulated by ERN1. Furthermore, results from *N. benthamiana* leaves, using *Epr3* short promoters and EMSA, indicate that both NIN and ERN1 control *Epr3* in the cortex (Fig. 7). Adding one more regulatory level, the fourfold increase in *Epr3* transcript levels observed in the *lhk1-1* mutant at 3 dpi and the increased epidermal expression of *Epr3:GUS* in *lhk1-1* roots suggests that at early time points perception of cytokinin by LHK1 is important not only for controlling *Nin* levels[43] but also for the feedback regulation of *Epr3* in the epidermal root hairs where infection threads are initiated (Fig. 7g). This feedback loop may involve ethylene as proposed for cytokinin regulation of infection thread formation[56].

Taken together, our results from the *Epr3* transcriptional activation in both symbiotic pathways initiated downstream of calcium spiking corroborate with the observed effects of *Epr3* mutations on infection. Thus, the results provide evidence for the role of this third LysM receptor during bacterial passage through the epidermis and subsequently during infection thread progression into the underlying cortical root tissues and nodule primordia.

## Methods

**Plant material and growth conditions.** *Lotus* ecotype Gifu[68] was used as the wild-type plant. The *epr3-9* and *epr3-10* alleles are single-nucleotide mutations, *epr3-11* is a LORE 1 retrotransposon insertion[57,69]. For nodulations assays at 21 and 28 °C plants were grown in growth chambers (Sanyo MLR-351) with day/night cycles of 16/8 h. Plants were inoculated with $OD_{600} = 0.02$ bacterial suspensions.

**Bacterial strains and growth conditions.** Wild-type *Mesorhizobium loti* strains R7A (refs 70,71) and MAFF303099 (ref. 72) along with the EPS mutant R7AexoB[57,58] were grown in TY/YMB medium at 28 °C. DsRed-tagged strains were used in microscopy experiments[57,58,73]. *Agrobacterium rhizogenes* strains AR12 or AR1193 (refs 74,75) were used for all hairy root transformation experiments[76]. *Agrobacterium tumefaciens* strain AGL1 was used in *N. benthamiana* infiltration experiments. The following concentrations of antibiotics were used for *M. loti* and *Agrobacterium* strains: ampicillin, 100 µg ml⁻¹; tetracycline, 2–10 µg ml⁻¹; gentamicin, 50 µg ml⁻¹; rifampicin, 100 µg ml⁻¹; spectinomycin, 100 µg ml⁻¹; and neomycin, 100 µg ml⁻¹ and carb, 100 µg ml⁻¹ for *A. tumefaciens* AGL1.

**Microscopy.** Nodules for light and TEM were embedded and sectioned[7]. Semi-thin nodule sections (5 µm) for light microscopy and ultra-thin (70 nm) sections for TEM were sliced using a Leica UCT ultramicrotome. Confocal microscopy of cortical infection threads was performed by hand-slicing 2-week-old nodules that were then stained with 0.04% Calcofluor White M2R (Fluorescent Brightener 28; Sigma). The cell wall and cortical infection threads were detected using a 410–490 nm filter, and *M. loti* DsRed were detected using a 587–665 nm

filter on a Zeiss LSM 710 confocal microscope. For *pEpr3:tYFP-NLS*, *pNfr1:tYFP-NLS* and *pNfr5:tYFP-NLS* analysis, roots and nodule sections were prepared by hand-sectioning and nuclear fluorescence was observed using a 491–535 nm filter on a Zeiss LSM 710 confocal microscope.

**GUS staining.** *Lotus* roots were stained with 0.5 mg ml⁻¹ 5-bromo-4-chloro-3-indolyl-β-D-glucuronic acid, 100 mM potassium phosphate buffer (pH7.0), 1 mM potassium ferricyanide, 1 mM potassium ferrocyanide and 0.1% Triton X-100. *Lotus* roots were incubated at 21 °C overnight.

**Promoter and expression constructs.** *pEpr3:tYFP-NLS* and *pEpr3:GUS* full-length promoters[57] were used. Shortened *Epr3* promoter regions (684 bp, 329 bp, 280 bp, 280 bp a mutated in ERN1-binding site, 280 bp a mutated in NIN1-binding site, 257 bp and 257 bp a mutated in NIN1-binding site) and the full-length *Nsp2* gene were amplified by PCR and recombined into *pDONR207* (Invitrogen) using the Gateway BP reaction (Invitrogen). These entry vectors were recombined into *pIV10:GW:GUS* and *pCAMBIA1300:Ljubiquitin promoter:GW*[77] destination vectors using Gateway LR reactions (Invitrogen). The *Nfr1* promoter (4,171 bp) and terminator (394 bp) regions, and for *Nfr5*, promoter (1,327 bp) and terminator (444 bp) have been used for building the *pNfr1:GUS*, *pNfr1:tYFP-NLS*, *pNfr5:GUS*, *pNfr5:tYFP-NLS* transcriptional reporter constructs. The resultant constructs were transformed into *A. rhizogenes* strain AR1193 and *A. tumefaciens* strain AGL1. See Supplementary Table 2 for primer sets.

**Epr3 induction analysis using N. benthamiana.** *A. tumefaciens* strain AGL1 transformed with appropriate vectors was grown in LB medium overnight. Cell pellets were collected by centrifugation and resuspended in 10 mM MgCl₂, 10 mM MES (pH 6.5) and 0.15 mM acetosyringone. Suspensions were diluted to: $OD_{600} = 0.1$ for *pEpr3:GUS*, $OD_{600} = 0.4$ for *pUbi:Nin*, *pUbi:Nsp2*, *p35S:Cyclops*, *p35S:Cyclops DD* and mixed with $OD_{600} = 0.05$ *A. tumefaciens* expressing the silencing suppressor p19. The mixed cultures were infiltrated into 4-week-old *N. benthamiana* leaves and 3 days later, leaves were collected and stained as described in the GUS-staining section at 37 °C overnight. After staining, leaves were decolorized using a 1:1:3 acetic acid:glycerol:ethanol solution at 65 °C twice for 1 h. GUS activity was quantified using 4-MUG[78] and fluorescence at 365 nm was measured on a Varian Cary Eclipse Fluorescence Spectrophotometer (Agilent).

**Quantitative PCR with reverse transcription gene expression analysis.** Nod factor and BAP were applied to roots and gene expression analysed[79]. For gene expression analysis in *nin-2* and *lhk1-1* mutants 10 plants were grown on ¼ B&D plates and inoculated with *M. loti* MAFF303099 or water (mock) at 14 or 3 days before collecting of roots. Three biological replicates were performed for each combination of mutant and time point. mRNA extraction, quality control checks and gene expression analysis using a Lightcycler480 instrument (Roche)[57].

**Protein expression and purification.** The open reading frame corresponding to full-length ERN1 and NIN (residues 520–878) were cloned into the pET30 vector and proteins were expressed in *E. coli* LOBSTR cells[80] (Kerafast, Boston, USA). Bacterial cultures were grown at 37 °C to an optical density (OD600) of ~0.6 and induced overnight with 0.2 mM isopropyl-β-D-thiogalactoside at 18 °C. Cells were opened by sonication in lysis buffer 1 for ERN1 (8 M urea, 25 mM Tris pH 7.8, 500 mM NaCl, 50 mM MgCl₂, 10% glycerol and 30 mM imidazole) and in lysis buffer 2 for NIN (500 mM NaCl, 25 mM Tris pH 7.8 and 30 mM imidazole), supernatants were cleared by centrifugation, loaded onto Ni-sepharose 6 FF affinity resins (GE Healthcare) and extensively washed in their respective lysis buffers. ERN1 was refolded on the Ni-resin by adding native buffer (25 mM Tris pH 7.8, 500 mM NaCl, 50 mM MgCl₂, 10% glycerol and 30 mM imidazole) following elution in the same buffer supplemented with 400 mM imidazole and NIN was eluted with lysis buffer 2 containing 400 mM imidazole.

**Electrophoretic mobility shift assay.** ERN1 and NIN were concentrated to 0.75 and 1.0 mg ml⁻¹, respectively, and mixed with (5(6)-Carboxyfluorescein) (FAM)-labelled promoter DNA segments (Supplementary Table 2). DNA probes (final volum 25 nM) and 2 µl protein were incubated in binding buffer (25 mM Tris-HCl pH8.0, 80 mM NaCl, 35 mM KCl, 5 mM MgCl₂, 10% (v/v) glycerol, 1 mM dithiothreitol and 0.1% (v/v) Triton100) for 30 min at 30 °C for ERN1 and 37 °C for NIN. These samples were subsequently separated in polyacrylamide gels (Criterion TGX Precast Gels; Bio-Rad) with 0.5× TBE buffer at 80 V for 120 min. FAM fluorescence was detected using a Typhoon Trio (Amersham)

**Data availability.** The EPR3 sequence is available in Genebank, accession no. BAI79284.1. Other informational data are available within the manuscript file and its Supplementary Files or are available from the corresponding author upon request.

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

## Acknowledgements
We thank Makoto Hayashi for the NIN construct and Martin Parniske for the CYCLOPS and CYCLOPS-DD constructs made available for this work, Susanne Buchholdt and Mette Hofmann Asmussen for technical assistance and Finn Pedersen for taking care of the plants. This work was supported by the Danish National Research Foundation grant no. DNRF79 and the ERC Advanced Grant 268523.

## Author contributions
Y.K.: expression studies and regulation of *Epr3*. M.W.N. and S.K.: expression studies and phenotyping. E.K.J.: sectioning and TE microscopy. S.R.R., W.F. and L.H.M.: construction of reporter genes. A.B.H.: quantitative PCR with reverse transcription. K.R.A.: protein purification. S.R. and J.S. conceived and coordinated experiments. J.S. and S.R. wrote the manuscript with input from Y.K. and S.K.

## Additional information

**Competing financial interests:** The authors declare no competing financial interests.

