## [Peer Review File · Nature Communications]

Reviewers' comments:

Reviewer #1 (Remarks to the Author):

Rhizobium-derived exopolysaccharides (EPS) are essential for the development of infected root nodules. Prof Stougaard and colleagues identified EPR3 LysM-containing receptor involved in sensing of bacterial EPS by a forward genetic approach in the previous work (Kawaharada et al., Nature, 2015).

This submitted manuscript further characterized *epr3* mutant phenotype and investigated genetic pathway/factors involved in Epr3 transcriptional regulation. Authors proposed a role for EPR3 in initiation of infection threads and release of rhizobia from infection threads. This notion is consistent with microscopic data (Fig. 1-3), and is a novel point of this manuscript.

In the second part, a lot of work was performed for Epr3 expression analysis. Firstly, kinetics of the Epr3 promoter activation investigated using an YFP-NLS reporter was consistent with EPR3 function deduced from the loss-of-function phenotype (Fig. 4, 5). Secondary, requirement of Nod factor receptors and of common SYM factors for Epr3 expression in response to rhizobial inoculation was shown by genetic analysis and by comparison with expression patterns of Nod factor receptor genes (Fig. 5, 6). These data reinforced the previous notion of two-step recognition signaling mediated Nod factors and EPS to control rhizobial infection. Thirdly, authors found two transcription factors primarily regulating Epr3 expression. ERN1 is essential for Epr3 activation in infected root hairs, and NIN is involved in expression in nodule primordia (Fig. 6, 7). Point mutations in a putative ERN1-binding site and in a NIN-binding site on the promoter (Fig 7c) indicated that these sites contribute to Epr3 transcriptional regulation in the root epidermis and nodule primordia, respectively. These data confer a novelty on the manuscript, however, not sufficient for clearly indicating that ERN1 and NIN target the Epr3 promoter. To clarify this point, gel shift analysis is essential.

Authors concluded that Epr3 expression in the root epidermis and cortex is regulated by different pathways downstream of common SYM. However, GUS expression data of mutant versions of 280 bp and 257 bp Epr3 promoters (Fig. 7c) does not exclude a possibility that ERN1 is also involved in the transcriptional regulation in primordia. Medicago ERN1 expresses in both infected root hairs and cortical cells. The 280 bp Epr3 promoter fragment with mutations in putative NIN binding sites will give information.

Expression data of *N. benthamiana* transient system (Fig 7d) are incredible, because reporter expression levels are altered by infection efficiencies of *Agrobacterium*. GUS expression levels should be quantitatively analyzed. Normalization with a reference control that is co-introduced with the reporter gene is necessary.

Although many data of expression analyses have been presented, there are no data functional relationship between ERN1/NIN and EPR3 to regulate host legume-bacterial compatibility. This point is critically important to explain how the compatibility is regulated at transcriptional levels.

Minor points

P6, L22; Figure number is not indicated.

Figure 4; Temporal expression data is not informative, because Epr3 expression in the epidermis in response to bacterial infection has already shown in the previous paper and these data overlap with Fig. 5. This figure should move to supplemental information, if it is dispensable for interpreting EPR3 function.

Figure 7b; Please fix mutarion.

Supplementary table 1

GUS expression in the cortex is always detected in mutants that can produce nodule primordia. Mutants with no ability to produce nodule primordia should be indicated.

In supplementary material and method, *L. japonicus* mutants were not described except *epr3*. I suppose that *L. japonicus* *ern1-2* has not been reported anywhere so far.

Reviewer #2 (Remarks to the Author):

In the manuscript "Differential regulation of the Epr3 receptor coordinates membrane-restricted rhizobial colonization of root nodule primordia", Kawaharada et al reported the EPS receptor EPR3 is necessary for efficient intracellular infection threads. They also revealed the transcriptional regulation of Epr3, which is upregulated after Nod factor perception. This is a very timely study on the function and regulation of this novel host factor for EPS perception. It is suitable for publication after considering the following comments from this reviewer:

The most significant comments (1-3) concern the organisation of figures.

1. Fig. 1g really should be part of Fig. 2, because both are about nodule phenotypes
2. In current Fig. 2, the right two columns are most informative, yet the panels are too small. This reviewer had to magnify to 400% to see the details. Consider reducing the left most column to insets. Or at least adopt the proportion used for Supplementary Fig. 1.
3. In Fig. 7, the arrangement of Constructs 1-7 is counterintuitive. Should reverse the order. Could flip the diagram in 7a too.

Other comments:

4. Could the authors please elaborate why cytokinin has opposing effects on Epr3 expression? One can claim that in epidermis, CK signalling represses EPR3 expression. In the cortex/nodule, however, *snf2* mutation seems not to affect EPR3 expression, or may slightly repress EPR3 expression. This reviewer finds the connection with CK signalling rather tenuous. The heightened expression level of EPR3 in *lhk1* mutant seems merely a consequence of hyperinfection.

Also, did the authors make sure BAP treatment was effective? In other words, did CK

responsive genes get induced?

Minor errors/typos

5. Page 4 Line 4, "uncharacteristed" should be "uncharacterised".
6. Page 4 Lines 9-11, "A LRR-receptor...protein kinase" is not a full sentence.
7. Page 5 Line 17, "Lotus wild type" should be "wild type Lotus".
8. Page 6 Line 11, the full form of "dpi" should be spelled out the first time it appears, not on Page 12, Line 16.
9. Page 6 Line 22, "Fig. m-p" is missing figure number.
10. Page 7 Line 18, "in" should be "among".
11. Page 9 Line 18, "though" should be "through".
12. Page 10 Line 3, should start a new paragraph to break a 35-line block.
13. Page 10 Line 14, "previous" should be "previously".
14. Page 13 Line 5, "less" should be "fewer" to avoid misunderstanding these nodules as less pink (if indeed this is what the authors intend).
15. Page 14 Line 1, "intercllular" should be "intercellular".
16. Page 15 Line 5, "controls" should be "control".
17. Page 15 Line 6, "controls" should be "control".
18. Page 15 Line 7, "Cyclops and Ern1" should be "cyclops and ern1".
19. Page 17 Line 12, "White arrow indicates a transcellular infection thread" should be "White arrows indicate transcellular infection threads".
20. Page 17 Line 21, what does calcofluor staining do?
21. Page 28, where does the blue fluorescence in Fig. 5 come from?
22. Page 30, in Fig. 7b, the two instances of "mutarion" should be "mutation".

Reviewer #3 (Remarks to the Author):

The paper "Differential regulation of the Epr3 receptor coordinates membrane-restricted rhizobial colonization of root nodule primordia" by Kawaharada et al. describes the role of a plant receptor in detecting and responding to exopolysaccharides (EPS) from rhizobial symbionts. It has been known for several decades that exopolysaccharides synthesized by nitrogen fixing bacterial symbionts are required for normal infection and subsequent colonization of legume nodules. However, the exact roles played by the EPS have not been elucidated. So, this is a long standing and important question in the field.

This paper shows that the plant EPS receptor (Epr3) is involved in allowing normal progression of infection threads through root hairs and into the underlying cell layers. It is also shown to be involved in promoting the construction of infection threads across cortical cells and across cells in the infection zone of the developing nodule. The authors present some evidence that in the absence of the "normal" route of infection via root hairs, that an alternative, more primitive, route of infection is used - "crack infection" whereby bacterial cells grow in intercellular spaces between root or nodule cells and enter the plant cells proper through pockets or infection pegs.

In addition, the authors do a nice job of placing the receptor in context in terms of where in the signaling network it is needed and which other signaling proteins are required for its function. Lastly the authors present work showing that transcription of the receptor is controlled by at least 2 promoter elements and transcription factors that result in specific, and separate, transcription in the plant epidermis and in the plant cortex.

Overall the paper is certainly of interest to workers in the field and to researchers in symbiosis in general. The results are novel, and the work showing how Epr3 is situated in the nodulation signaling/transcriptional regulation network is very valuable. The validity and robustness of the conclusions is high, and the methods used are appropriate. Questions concerning interpretation, methods etc. are below:

1. The importance of the Epr3 receptor is overstated, especially in the discussion. It is referred to as being "essential" for progression of infection threads through the root hair and underlying cells. The term "essential" usually denotes a function that is, well, essential. That is, something fails to happen if the essential function is missing. This is not the case with Epr3 and infection. Infection thread progression is certainly altered, stopping, blebbing and then growing again. The threads DO grow normally for much of the distance down the root hair. This is shown in the Figure 1 photos and in the interpretive drawings in that figure. Transcellular threads are also made in the Epr3 mutant as is shown in the bar graphs in Figure 1. So again, there is an altered phenotype, but the receptor is not "essential" for the formation of transcellular threads. Same with the word "abrogation".

The authors should clarify where the 5 sections came from for the data in the bar graphs in Figure 1 - were they sequential or spaced throughout the nodule?

Along these lines, the phenotype of a bacterial EPS- mutant is quite a bit stronger than the Epr3 mutants (see supplementary Fig 6 as an example). This suggests that either Epr3 mutants retain some function, or there is a redundant receptor, or other function, that can provide some signaling that is missing in the Epr3 mutants. Discussion of this would be

useful.

2. Figure 3 shows sections through nodule tissue with red arrows indicating infection pegs; other aberrant structures shown as well. Because the the infection threads in nodule tissue can be at any orientation with respect to the plane of the section, it is very hard to determine if the structures called "infection pegs" are really that. For example, the peg highlighted in Fig 3f may be a cross section of a normal transcellular infection thread. That show in panel "n" may be an oblique section of the same. Overall, this figure does not contribute much to the paper.

3. Page 11 line 9: The authors state that cytokinins have opposing effects on Epr3 expression in epidermal and cortical cells. It was not clear how this conclusion was reached- perhaps they can clarify or reiterate the data, or logic, that supports this conclusion.

4. The signaling network leading to infection, nodulation, calcium spiking, EPS sensing, etc. is complicated, but fairly well known. A figure showing this network and the position of Epr3, its regulators and its downstream functions would be enormously useful, and help the reader better integrate all of the signal-transduction and transcriptional regulation information in the manuscript.

Minor things:

1. A sentence or two speculating on why intracellular threads stop growing and then restart more often in the Epr3 and EPS mutants would be interesting. Also-these events do happen in wild type infections, they are just less common. Are wild-type instances of this somehow tied to EPS production or sensing?
2. Why is wild type MAFF303099 used for Figure 4 (see page 7, line 18) when wild type R7A was used for the previous experiments?
3. n values are needed for Supp. Fig 5
4. In Supp Fig 6, do the y-axes show the average number of nodules per plant? They seem low. What do the error bars show?
5. What do the error bars in Fig 1 show?

Reviewer #4 (Remarks to the Author):

The manuscript follows on from a previous publication in Nature. Unfortunately there is very little novelty presented and very few advances documented in this new submission. The authors focus the work on the characterization of EPR3 function during cortical infection. They find that this mostly repeats what they previously observed in the epidermis, that EPR3 is necessary for infection thread development and is expressed in cells where infection threads occur. The fact that the cortex reflects what happens in the epidermis is not surprising. The most novel aspects of the work is the demonstration that NIN and ERN are

involved in the activation of EPR3. However, the proof for this is limited, no in vivo or in vitro binding is demonstrated. Overall, the novelty of the work is not particularly strong.

The data presented is generally of a high quality. The microscopic images are excellent. For improvement of the data presented I would suggest:

1. qPCR validation is necessary to support the EPR3-GUS work in the mutants
2. Controls for the Nicotiana work would be beneficial, for instance a demonstration that CYCLOPS DD can activate pNIN-GUS.
3. The data presented for ERN1 induction of pEPR3 is not very convincing. The induction looks very weak to me. This should be validated with quantification, not just GUS staining.
4. The NIN induction of pEPR3 is not validated by any other means and the pEPR3-GUS in the nin mutant suggests that win is not required. Analysis of later stages nin mutants should be able to validate the proposal that NIN regulates EPR3 in the cortex.
5. The evidence for NIN and ERN1 regulation of pEPR3 is very limited. The authors could mutate the predicted cis elements in the EPR3 promoter and demonstrate that these are required. The authors could also demonstrate that ERN1 and NIN can bind the pEPR3 promoter using EMSA and ChIP. As it stands I don't believe that the evidence presented is sufficient to support the claims made that these transcription factors regulate pEPR3.
6. The only convincing evidence for ERN and NIN induction of pEPR3 is in Nicotiana. Is this also the case in Lotus roots?

Reviewer #1 (Remarks to the Author):

Comment: Rhizobium-derived exopolysaccharides (EPS) are essential for the development of infected root nodules. Prof Stougaard and colleagues identified EPR3 LysM-containing receptor involved in sensing of bacterial EPS by a forward genetic approach in the previous work (Kawaharada et al., Nature, 2015).

This submitted manuscript further characterized *epr3* mutant phenotype and investigated genetic pathway/factors involved in Epr3 transcriptional regulation. Authors proposed a role for EPR3 in initiation of infection threads and release of rhizobia from infection threads. This notion is consistent with microscopic data (Fig. 1-3), and is a novel point of this manuscript.

In the second part, a lot of work was performed for Epr3 expression analysis. Firstly, kinetics of the Epr3 promoter activation investigated using an YFP-NLS reporter was consistent with EPR3 function deduced from the loss-of-function phenotype (Fig. 4, 5). Secondary, requirement of Nod factor receptors and of common SYM factors for Epr3 expression in response to rhizobial inoculation was shown by genetic analysis and by comparison with expression patterns of Nod factor receptor genes (Fig. 5, 6). These data reinforced the previous notion of two-step recognition signaling mediated Nod factors and EPS to control rhizobial infection. Thirdly, authors found two transcription factors primarily regulating Epr3 expression. ERN1 is essential for Epr3 activation in infected root hairs, and NIN is involved in expression in nodule primordia (Fig. 6, 7). Point mutations in a putative ERN1-binding site and in a NIN-binding site on the promoter (Fig 7c) indicated that these sites contribute to

Epr3 transcriptional regulation in the root epidermis and nodule primordia, respectively. These data confer a novelty on the manuscript, however, not sufficient for clearly indicating that ERN1 and NIN target the Epr3 promoter. To clarify this point, gel shift analysis is essential.

Response: Electrophoretic mobility shift assays (EMSA)s are now included in the revised Figure 7 b,f and the following sentences added to the manuscript page 11 *“The role of the predicted ERN1 and NIN binding sites of the 280 bp Epr3 promoter segment (Fig. 7b) was further investigated in electrophoretic mobility shift assay (EMSA) (Fig. 7f). Full length ERN1 and truncated NIN (residues 520-878) proteins were expressed in E. coli and binding of DNA probes encompassing the corresponding binding sites determined as mobility shifts (Fig. 7b, f and Table S2). Mobility shifts were observed for both proteins and binding of DNA fragments carrying mutations found to impair epidermal or cortical expression in planta was eliminated or reduced (Fig. 7f).”*

Comment: Authors concluded that Epr3 expression in the root epidermis and cortex is regulated by different pathways downstream of common SYM. However, GUS expression data of mutant versions of 280 bp and 257 bp Epr3 promoters (Fig. 7c) does not exclude a possibility that ERN1 is also involved in the transcriptional regulation in primordia.

Medicago ERN1 expresses in both infected root hairs and cortical cells. The 280 bp Epr3 promoter fragment with mutations in putative NIN binding sites will give information.

Response: Analysis of the 280 bp Epr3 promoter with mutations in the NIN binding site is now included in Figure 7 and the following added to the manuscript page 11 *“In contrast, no detectable change in epidermal expression was observed after mutation of a sequence corresponding to a previously identified NIN binding site^{40,63} in the 280 promoter (mutation 2). However, mutation of this NIN binding site in the 257 bp promoter eliminates the cortical expression (Fig. 7b, c) suggesting that ERN1 can still activate the Epr3 280 promoter lacking the NIN binding site (mutation 2) in the cortex.*

Comment: Expression data of *N. benthamiana* transient system (Fig 7d) are incredible, because reporter expression levels are altered by infection efficiencies of *Agrobacterium*. GUS expression levels should be quantitatively analyzed. Normalization with a reference control that is co-introduced with the reporter gene is necessary.

Response: Expression in *N. benthamiana* has been quantified and this information is now included in the revised Figure 7e.

Comment: Although many data of expression analyses have been presented, there are no data functional relationship between ERN1/NIN and EPR3 to regulate host legume-bacterial compatibility. This point is critically important to explain how the compatibility is regulated at transcriptional levels.

Response: Understanding the *Epr3* dependent response to compatible and incompatible bacteria is important but clearly outside the scope of this manuscript. Our ongoing analysis of the early response down-stream of EPR3 points towards posttranscriptional regulation rather than transcriptional changes.

Comment: Minor points

P6, L22; Figure number is not indicated.

Response: corrected

Comment: Figure 4; Temporal expression data is not informative, because *Epr3* expression in the epidermis in response to bacterial infection has already shown in the previous paper and these data overlap with Fig. 5. This figure should move to supplemental information, if it is dispensable for interpreting EPR3 function.

Response: Figure 4 incorporates the temporal and spatial expression in root hairs and nodule primordia at a whole root level. We find this Figure a very illustrative description integrating *Epr3* expression in the infection and nodule developmental events. We believe it is of value for non-expert readers and has left it in.

Comment: Figure 7b; Please fix mutation.

Response: Corrected in the revised Figure 7

Comment: Supplementary table 1

GUS expression in the cortex is always detected in mutants that can produce nodule primordia. Mutants with no ability to produce nodule primordia should be indicated.

Response: Table S1 revised as suggested

Comment: In supplementary material and method, *L. japonicus* mutants were not described except *epr3*. I suppose that *L. japonicus ern1-2* has not been reported anywhere so far.

Response: The *ern1-2* mutants is described in a separate manuscript

Reviewer #2 (Remarks to the Author):

In the manuscript "Differential regulation of the Epr3 receptor coordinates membrane-restricted rhizobial colonization of root nodule primordia", Kawaharada et al reported the EPS receptor EPR3 is necessary for efficient intracellular infection threads. They also revealed the transcriptional regulation of Epr3, which is upregulated after Nod factor perception. This is a very timely study on the function and regulation of this novel host factor for EPS perception. It is suitable for publication after considering the following comments from this reviewer:

The most significant comments (1-3) concern the organisation of figures.

Comment: 1. Fig. 1g really should be part of Fig. 2, because both are about nodule phenotypes

Response: Fig 1g moved to Figure 2.

Comment: 2. In current Fig. 2, the right two columns are most informative, yet the panels are too small. This reviewer had to magnify to 400% to see the details. Consider reducing the left most column to insets. Or at least adopt the proportion used for Supplementary Fig. 1.

Response: Proportion changed to enhance visibility of the two columns to the right.

Comment: 3. In Fig. 7, the arrangement of Constructs 1-7 is counterintuitive. Should reverse the order. Could flip the diagram in 7a too.

Response: Order of construct 1- 7 reversed as suggested.

Other comments:

Comment: 4. Could the authors please elaborate why cytokinin has opposing effects on Epr3 expression? One can claim that in epidermis, CK signalling represses EPR3 expression. In the cortex/nodule, however, *snf2* mutation seems not to affect EPR3 expression, or may slightly repress EPR3 expression. This reviewer finds the connection with CK signalling rather tenuous. The heightened expression level of EPR3 in *lhk1* mutant seems merely a consequence of hyperinfection.

Also, did the authors make sure BAP treatment was effective? In other words, did CK responsive genes get induced?

Response: Point taken. The cytokinin response has been clarified and page 12 now reads "*This indicates that cytokinin signaling is, directly or indirectly, involved in repression of epidermal Epr3 expression and correlates the Epr3 expression level to the frequency of infection thread progression known to be increased in lhk1-1 mutants*"⁴⁴.

Our detailed analyses of the onset of Epr3 transcription in various symbiotic mutants revealed that the activation of the EPS recognition step has a differential genetic requirement in the epidermal cell layer, compared to the root cortex where nodule primordia emerge. The increased Epr3 expression in lhk1-1 mutants and induction of Epr3 in primordia of snf2 mutants suggest that cytokinin signalling have opposing effects on Epr3 expression in epidermal and cortical cells."

For BAP treatments standard condition used for cytokinin studies in mutants and wt plants were used , see Reid et al 2016 Plant Phys, 170, 1060

Comment: Minor errors/typos

5. Page 4 Line 4, "uncharacteristed" should be "uncharacterised".

6. Page 4 Lines 9-11, "A LRR-receptor...protein kinase" is not a full sentence.
7. Page 5 Line 17, "Lotus wild type" should be "wild type Lotus".
8. Page 6 Line 11, the full form of "dpi" should be spelled out the first time it appears, not on Page 12, Line 16.
9. Page 6 Line 22, "Fig. m-p" is missing figure number.
10. Page 7 Line 18, "in" should be "among".
11. Page 9 Line 18, "though" should be "through".
12. Page 10 Line 3, should start a new paragraph to break a 35-line block.
13. Page 10 Line 14, "previous" should be "previously".
14. Page 13 Line 5, "less" should be "fewer" to avoid misunderstanding these nodules as less pink (if indeed this is what the authors intend).
15. Page 14 Line 1, "intercllular" should be "intercellular".
16. Page 15 Line 5, "controls" should be "control".
17. Page 15 Line 6, "controls" should be "control".
18. Page 15 Line 7, "Cyclops and Ern1" should be "cyclops and ern1".
19. Page 17 Line 12, "White arrow indicates a transcellular infection thread" should be "White arrows indicate transcellular infection threads".
Response: Comments 5 to 19 corrected in manuscript

20. Page 17 Line 21, what does calcofluor staining do?

Response: Information included and sentence changed to "*At 10 to 14 dpi nodules were sectioned by hand and cell walls stained with 0.04% calcofluor.*"

21. Page 28, where does the blue fluorescence in Fig. 5 come from?

Response: The blue fluorescence is autofluorescence detected using a filter setting described in supplementary methods.

22. Page 30, in Fig. 7b, the two instances of "mutarion" should be "mutation".

Response: Corrected in the revised Figure 7

Reviewer #3 (Remarks to the Author):

The paper "Differential regulation of the Epr3 receptor coordinates membrane-restricted rhizobial colonization of root nodule primordia" by Kawaharada et al. describes the role of a plant receptor in detecting and responding to exopolysaccharides (EPS) from rhizobial symbionts. It has been known for several decades that exopolysaccharides synthesized by nitrogen fixing bacterial symbionts are required for normal infection and subsequent colonization of legume nodules. However, the exact roles played by the EPS have not been elucidated. So, this is a long standing and important question in the field.

This paper shows that the plant EPS receptor (Epr3) is involved in allowing normal progression of infection threads through root hairs and into the underlying cell layers. It is also shown to be involved in promoting the construction of infection threads across cortical cells and across cells in the infection zone of the developing nodule. The authors present some evidence that in the absence of the "normal" route of infection via root hairs, that an alternative, more primitive, route of infection is used-"crack infection" whereby bacterial grow in intercellular spaces between root or nodule cells and enter the plant cells proper through pockets or infection pegs.

In addition, the authors do a nice job of placing the receptor in context in terms of where in the signaling network it is needed and which other signaling proteins are required for its function. Lastly the authors present work showing that transcription of the receptor is controlled by at least 2 promoter elements and transcription factors that result in specific, and separate, transcription in the plant epidermis and in the plant cortex. Overall the paper is certainly of interest to workers in the field and to researchers in symbiosis in general. The results are novel, and the work showing how Epr3 is situated in the nodulation signaling/transcriptional regulation network is very valuable. The validity and robustness of the conclusions is high, and the methods used are appropriate. Questions concerning interpretation, methods etc. are below:

Comment: 1. The importance of the Epr3 receptor is overstated, especially in the discussion. It is referred to as being "essential" for progression of infection threads through the root hair and underlying cells. The term "essential" usually denotes a function that is, well, essential. That is, something fails to happen if the essential function is missing. This is not the case with Epr3 and infection. Infection thread progression is certainly altered, stopping, blebbing and then growing again. The threads DO grow normally for much of the distance down the root hair. This is shown in the Figure 1 photos and in the interpretive drawings in that figure. Transcellular threads are also made in the Epr3 mutant as is shown in the bar graphs in Fig 1. So again, there is an altered phenotype, but the receptor is not "essential" for the formation of transcellular threads. Same with the word "abrogation".

Response: Essential has been replaced by "promotes" or "advance" progression of infection threads and abrogated by "impaired"

Comment: The authors should clarify where the 5 sections came from for the data in the bar graphs in Figure 1-were they sequential or spaced throughout the nodule

Response: Information added and legend now reads "*Number of cortical infection threads were counted in 5 randomly selected sections of 5 nodules from each combination*"

Comment: Along these lines, the phenotype of a bacterial EPS- mutant quite a bit stronger than the Epr3 mutants (see supplementary Fig 6 as an example). This suggests that either Epr3 mutants retain some function, or there is a redundant receptor, or other

function, that can provide some signaling that is missing in the Epr3 mutants. Discussion of this would be useful.

Response: The following sentence was added to the discussion page 14 "*Although a passive physicochemical contribution from EPS can not be excluded, the phenotypic differences between epr3 mutants and wild type plants inoculated with R7AexoB (Supplemental Figure 6) may suggest that EPR3 have a co-receptor that is partially functional in the absence of EPR3.*"

Comment: 2. Figure 3 shows sections through nodule tissue with red arrows indicating infection pegs; other aberrant structures shown as well. Because the the infection threads in nodule tissue can be at any orientation with respect to the plane of the section, it is very hard to determine if the structures called "infection pegs" are really that. For example, the peg highlighted in Fig 3f may be a cross section of a normal transcellular infection thread. That show in panel "n" may be an oblique section of the same. Overall, this figure does not contribute much to the paper.

Response: We can only report what we see. The structures called pegs were not observed in wild type plants inoculated with wild type *M. loti*. Previous studies Madsen et al Nature Communications 2010 1:10, see Fig 3 j,k,i, showed that such protrusions were associated with accumulation of bacteria outside the cells. Here we show images from the inside which corresponds well to the pegs and infection threads shown in Fig. 2.

Comment: 3. Page 11 line 9: The authors state that cytokinins have opposing effects on Epr3 expression in epidermal and cortical cells. It was not clear how this conclusion was reached-perhaps they can clarify or reiterate the data, or logic, that supports this conclusion.

Response: See response to reviewer 2.

Comment: 4. The signaling network leading to infection, nodulation, calcium spiking, EPS sensing, etc. is complicated, but fairly well known. A figure showing this network and the position of Epr3, its regulators and its downstream functions would be enormously useful, and help the reader better integrate all of the signal-transduction and transcriptional regulation information in the manuscript.

Response: We have added a working model in Figure 7g.

Minor things:

Comment: 1. A sentence or two speculating on why intracellular threads stop growing and then restart more often in the Epr3 and EPS mutants would be interesting. Also- these events do happen in wild type infections, they are just less common. Are wild-type instances of this somehow tied to EPS production or sensing?

Response: Interesting questions unfortunately we do not at this point have the refined analysis necessary to come up with a meaningful speculation. A sentence on the possible passive physicochemical role of EPS and a possible existence of a co-receptor was added to the discussion see above and that is about as far as we can go.

Comment: 2. Why is wild type MAFF303099 used for Figure 4 (see page 7, line 18) when wild type R7A was used for the previous experiments?

Response: The MAFF303099 DsRed has a stronger DsRed signal and this was necessary to capture the bacterial presence in these whole mounts.

Comment: 3. n values are needed for Supp. Fig 5

Response: A detailed description of the analyses presented in Supp. Fig 5, including the n values are presented in the Supplemental experimental procedure. "*Expression analysis of Nod factor and BAP-treated roots was conducted as described previously*¹⁶. For gene expression analysis in *nin-2* and *lhk1-1* mutants 10 plants were grown on 1/4 B&D plates and inoculated with *M. loti* MAFF303099 or water (mock) at 14 or 3 days before harvesting of roots. Three biological replicates were performed for each combination of mutant and time point."

Comment: 4. In Supp Fig 6, do the y-axes show the average number of nodules per plant? They seem low. What do the error bars show?

Response: Missing information added to Figure and legend. Error bars are SE.

Comment: 5. What do the error bars in Fig 1 show?

Response: Missing information added to legend. Error bars are SE.

Reviewer #4 (Remarks to the Author):

Comment: The manuscript follows on from a previous publication in Nature. Unfortunately there is very little novelty presented and very few advances documented in this new submission. The authors focus the work on the characterization of EPR3 function during cortical infection. They find that this mostly repeats what they previously observed in the epidermis, that EPR3 is necessary for infection thread development and is expressed in cells where infection threads occur. The fact that the cortex reflects what happens in the epidermis is not surprising. The most novel aspects of the work is the demonstration that NIN and ERN are involved in the activation of EPR3. However, the proof for this is limited, no in vivo or in vitro binding is demonstrated. Overall, the novelty of the work is not particularly strong.

Response: We thank the reviewer for pointing out that the intricacies of infection thread formation and progression in epidermis, cortical tissues and nodule primordia was not well explained and the novelty of our findings therefore not so obvious. Infection thread progression is easily mistaken for a form of hyphal growth and not a process that has to be initiated and controlled at each cell passage. Infection threads are initiated and propagated by the individual root cortex cells in a cell-autonomous process that is distinctly different from the first initiation in curled root hairs. Observations on requirement for *Epr3* in the cell-to-cell passage of the advancing infection thread described in this manuscript, does therefore not repeat previous single cell observations in the epidermis, as stated. Such cell-to-cell passage necessary for cortical invasion and final endocytosis could obviously not have been derived from our previous study in individual root hair cells. We have clarified the infection thread process in the introduction and highlighted the novelty by comparing the *epr3* mutant phenotype to

previously isolated mutants that show a difference in infection thread formation/progression in root hairs compared to infection thread progression in the cortex.

The following description of the infection thread process was added to the introduction on page 5 *"The biochemical pathway and genetic network subsequently involved in progressing infection threads into the root cortex is virtually undescribed and our understanding is mainly based on imaging. Individual plant cells initiates infection thread formation at the interface of the cell above and extend the infection thread to the interface of the cell below. Analysis of electron micrographs suggest that fusion of the infection thread wall at the site of entry into the lower cell precedes cell wall degradation and re-initiation of the infection tread in the lower cell⁵⁶. This iterated cell-autonomous process, which appears to differ from the initial infection chamber formation, advances the infection thread and by an unknown mechanisms branching occurs in the nodule primordium before bacteria are released from the infection thread into the plant cell. Infection thread progression is synchronized and coupled to the development of primordia in a highly regulated process that has not yet been uncovered. An example is the abortion of most infection threads already in the epidermis. This lead to the notion of an epidermal-cortical barrier where cytokinin signalling mediates repression⁵⁷ while endoreduplication promotes reinitiation^{46,47} suggesting that infection thread progression is regulated at each cell passage. Calcium oscillations observed in the plant cell just ahead of the growing infection thread suggest that Nod factor perception and CCaMK activation is involved but components of a regulatory mechanism controlling the cell autonomous advance of infection threads have not yet been identified."*

The following comparison of the regulatory *epr3* mutant phenotype and mutants impaired in infection thread development was added to the discussion on page 15 *"Such a regulatory role is supported by the different phenotype of *epr3* mutants compared to other infection thread mutants. In most of these developmental mutants, infection threads are arrested in the root hairs or misguided at the cortical boundary and little or no infection of primordia occurs^{21-23,33,46,47,49,51-54}."*

See response to reviewer 1 for our response to the comments on the Epr3 activation process.

The data presented is generally of a high quality. The microscopic images are excellent. For improvement of the data presented I would suggest:

Comment: 1. qPCR validation is necessary to support the EPR3-GUS work in the mutants

Response: The promoter studies provided clear plus minus results and the key observations have been verified in subsequent promoter deletion/mutation studies and gel shifts, see Figure 7.

Comment: 2. Controls for the Nicotiana work would be beneficial, for instance a demonstration that CYCLOPS DD can activate pNIN-GUS.

Response: Figure 7 was updated and now includes quantitative data. See response to reviewer 1

Comment: 3. The data presented for ERN1 induction of pEPR3 is not very convincing. The induction looks very weak to me. This should be validated with quantification, not just GUS staining.

Response: Figure 7 was updated and now includes quantitative data. See response to reviewer 1

Comment: 4. The NIN induction of pEPR3 is not validated by any other means and the pEPR3-GUS in the *nin* mutant suggests that *win* is not required. Analysis of later stages *nin* mutants should be able to validate the proposal that NIN regulates EPR3 in the cortex.

Response: We agree that *Nin* is not required in the epidermis and this is what we present. *Nin* mutants are impaired in the organogenic process and do not develop nodules or nodule primordia. There are thus no later stages to analyse. *Nin* involvement is supported by transient expression studies in *Nicotiana*, pEpr3-GUS studies in *Lotus* roots and analysis of binding sites in new Fig 7 showing that mutation of the site in the 257 promoter eliminate/reduces cortical *Epr3* expression.

Comments: 5. The evidence for NIN and ERN1 regulation of pEPR3 is very limited. The authors could mutate the predicted cis elements in the EPR3 promoter and demonstrate that these are required. The authors could also demonstrate that ERN1 and NIN can bind the pEPR3 promoter using EMSA and CHIP. As it stands I don't believe that the evidence presented is sufficient to support the claims made that these transcription factors regulate pEPR3.

Response: Results from the suggested experiments data was included in Figure 7. See comment to reviewer 1.

Comment: 6. The only convincing evidence for ERN and NIN induction of pEPR3 is in *Nicotiana*. Is this also the case in *Lotus* roots?

Response: The reviewer appears to have overlooked the analysis of *ern1* and *nin* mutants of *Lotus* in Fig 6 and Table S1. To clarify, we have in Figure 7 added gel shift analyses and additional analyses of ERN1 and NIN binding site mutations in the *Epr3* promoter transformed into *Lotus* roots.

Reviewers' comments:

Reviewer #1 (Remarks to the Author):

Authors improved Fig.7 panel c, and added new data. The gel shift analysis clearly showed that ERN1 recognized the putative binding site on the Epr3 promoter. This data is consistent with GUS expression in the panel c. However, the truncated NIN protein still bound with a probe after substitution of important nucleotides in putative NIN-binding site despite of no GUS expression from the 257 bp promoter with the same mutations in the cortex. These data are not clear. Further, the quantitative GUS expression assay in tobacco has not been normalized by a reference gene.

When the putative NIN-binding site in the 280 bp promoter was mutated, GUS expression was detected in both epidermis and developing nodules in panel c. The important novelty in this manuscript is Epr3 expression in the epidermis and cortex is regulated by different transcription factors. This data weakened the evidence supporting the proposed concept, but is consistent with ERN1 expression sites in Medicago roots. In addition, it has been known that NIN expresses in both epidermis and cortex, and that NIN target genes identified so far express in both cell layers, however, your data have shown that NIN activates Epr3 only in the cortex. Although this point may be a specific feature of the Epr3 gene, your promoter deletion experiments have not fully excluded a possibility that other NIN-binding sites in the Epr3 promoter contribute to Epr3 expression in the epidermis as well as the ERN1-binding site.

Reviewer #2 (Remarks to the Author):

I have reviewed the initial submission of this manuscript, and have recommended accepting the manuscript upon minor revisions. The revision has satisfactorily address all my questions from the first round of review. In addition, the authors have included additional experimental data to substantially improve Fig. 7. The EMSA results demonstrated direct binding of ERN1 and NIN proteins on the EPR3 promoter at the identified cis-elements. This data adds strong support to the model proposed at the end of the manuscript.

Now that this manuscript is, in my opinion, closer to the publication standards of this journal, I have subjected the revision to closer scrutiny. I hope the authors find the additional recommendations reasonable.

1. Fig. 2 legend (Page 18 Lines 18-19 and 23-24): The description of white arrows, red arrows, yellow arrows, and red "stars" (it should be called an "asterisk") should be together.
2. Fig. 3 legend (Page 19 Line 8) and other figure legends throughout the manuscript: Should consistently say "scale bars", rather than just "bars".
3. Fig. 4 legend (Page 19 Line 15): There is only one yellow arrow.

4. Fig. 7d. The authors can consider rotating/tilting the labels, so that they all fit in the same row.

5. The authors must pay attention to noun-verb agreement. Such errors permeate the text. The following are ones that I noticed: Page 2 Line 4; Page 4 Line 4; Page 5 Lines 8, 10, and 18; Page 6 Lines 9 and 14; Page 16 Lines 15 and 18; and Supplementary Figure 2 legend Line 6.

Reviewer #3 (Remarks to the Author):

The authors have done a very thorough job of addressing the concerns raised by the viewers, as a result the paper is much stronger than it was previously. The addition of gel-shift assays, new GUS reporter constructs and a myriad of other changes helps with the technical short-comings of the previous manuscript. The explanations of the manuscripts novelty and importance helps as well and removes confusion about how this paper differs from previous work from the PI's laboratory. Progression of infection threads through the cortex is indeed an altogether different process than progression through root hairs. It was in no way obvious that EPS would have a role in maintaining IT progression through the cortex and developing nodule, rather than in just the root-hair/epidermis alone.

Reviewer #4 (Remarks to the Author):

My original concerns with the first submission were that it lacked much novelty relative to the previous paper published in Nature. It is not surprising to me that the mechanisms that are required for initiation of infection threads and those that are required for progression are conserved. However, the authors also document the transcriptional regulation of *EPR3* and how this is associated with the progression of infection. For me this is a much more interesting aspect of the work, and the authors have now included additional evidence that makes a much stronger case for this. The authors demonstrate that *ERN* and *NIN* both regulate *EPR3* expression and apparently do so in different tissues. It is striking to me that the *epr3* mutant phenotype is very similar to that described for *ern*: swelling and bulging of infection threads. This similarity in mutant phenotypes is not mentioned in the manuscript and may be good to add. It would also be very interesting to see if over expression of *EPR3* compensated for the *ern* mutation, although I don't believe that this is essential for the current manuscript.

Overall the authors have provided a much stronger case for the transcriptional control of *EPR3* and how this appears to be important for the progression of infection threads. We know that *ERN* and *NIN* are necessary for infection thread formation, but the reasons why have been elusive. This report forms some of the first insights into what these transcription

factors are regulating during infection. This is a novel and interesting finding.

Reviewer #1 (Remarks to the Author):

Comment: Authors improved Fig.7 panel c, and added new data. The gel shift analysis clearly showed that ERN1 recognized the putative binding site on the Epr3 promoter. This data is consistent with GUS expression in the panel c. However, the truncated NIN protein still bound with a probe after substitution of important nucleotides in putative NIN-binding site despite of no GUS expression from the 257 bp promoter with the same mutations in the cortex. These data are not clear. Further, the quantitative GUS expression assay in tobacco has not been normalized by a reference gene.

When the putative NIN-binding site in the 280 bp promoter was mutated, GUS expression was detected in both epidermis and developing nodules in panel c. The important novelty in this manuscript is Epr3 expression in the epidermis and cortex is regulated by different transcription factors. This data weakened the evidence supporting the proposed concept, but is consistent with ERN1 expression sites in Medicago roots. In addition, it has been known that NIN expresses in both epidermis and cortex, and that NIN target genes identified so far express in both cell layers, however, your data have shown that NIN activates Epr3 only in the cortex. Although this point may be a specific feature of the Epr3 gene, your promoter deletion experiments have not fully excluded a possibility that other NIN-binding sites in the Epr3 promoter contribute to Epr3 expression in the epidermis as well as the ERN1-binding site.

Response: We thank the reviewer for support of our science and have addressed the reviewer's comments. Reviewer 1 suggests that the role of NIN in the epidermis and the importance of the NIN1 binding site in the 257 bp promoter remains unclear. In response we now present our conclusions from the promoter analysis more precisely and use a more specific terminology to clarify our results and avoid possible misinterpretations. The contribution of the putative NIN2 and NIN3 binding sites, which were accepted were previously not sufficiently described, have now also been addressed. Regarding the binding of NIN to the mutated NIN1 binding site we refer to Figure 7 f showing a weaker NIN binding to this mutated site and the corresponding text. Below, the text changes are highlighted in blue.

Page 10. “Further evidence for *differential* regulation of Epr3 expression in the epidermal and cortical primordia was obtained from *nena*, *cyclops*, *nsp1* and *ern1* mutants, which eventually initiate the organogenesis program in the cortical layers leading to nodule primordia formation, and from the *snf1* mutant (CCaMK gain of function) (Supplementary Fig. 4, and Supplementary Table 1). In these mutants, the Epr3 promoter was active in the emerging nodule primordia. We then investigated whether promoter sequences *mediating* epidermal and cortical Epr3 expression could be delimited in transgenic *Gifu* roots using a set of promoter deletions. In these *Gifu* roots, the 1019 bp⁵⁸, the 684 bp, the 329 bp and the 280 bp promoters induced GUS expression in both the epidermis and nodule primordia. However, further deletion of promoter sequences to 257 bp eliminated epidermal expression while activation in nodule primordia was still observed (Fig. 7a, c). This analysis positions a regulatory promoter element *sufficient* for epidermal expression between 280 and 257 bp and an element regulating primordial expression within the 257 bp promoter.”

Page 13. “These results suggest that ERN1 is essential for Epr3 expression in both epidermal and cortical cells and delimits an ERN1 binding site mediating epidermal expression of Epr3 in the epidermis. NIN is essential for induction of Epr3 expression in cortical cells (Fig. 7g) and a NIN1 binding site sufficient for cortical cell expression was delimited in the Epr3 promoter. The deletion study did not reveal a major contribution from the putative NIN2 and NIN3 binding sites in the Epr3 promoter. Taken together our results suggest that ERN1 is sufficient for epidermal Epr3 expression while NIN appears to be dependent on ERN1 for any epidermal regulation of Epr3 that may not have been uncovered by our promoter analysis.”

Page 21 legend for Fig 7(g). “Model of Epr3 induction. In the epidermis, ERN1 is sufficient for Epr3 expression down-stream of CCaMK, CYCLOPS and NSP2. In the cortical primordia, both NIN and ERN1 induce Epr3 expression directly down-stream of CCaMK, NSP2 and the LHK1 cytokinin receptor. Cytokinin signaling

through LHK1 has a negative effect on epidermal Epr3 expression possibly involving the mechanism regulating infection thread formation⁴⁴."

Reviewer #2 (Remarks to the Author):

Comment: I have reviewed the initial submission of this manuscript, and have recommended accepting the manuscript upon minor revisions. The revision has satisfactorily address all my questions from the first round of review. In addition, the authors have included additional experimental data to substantially improve Fig. 7. The EMSA results demonstrated direct binding of ERN1 and NIN proteins on the EPR3 promoter at the identified cis-elements. This data adds strong support to the model proposed at the end of the manuscript. Now that this manuscript is, in my opinion, closer to the publication standards of this journal, I have subjected the revision to closer scrutiny. I hope the authors find the additional recommendations reasonable.

Response: We thank the reviewer for support of our science

Comment: 1. Fig. 2 legend (Page 18 Lines 18-19 and 23-24): The description of white arrows, red arrows, yellow arrows, and red "stars" (it should be called an "asterisk") should be together.

Response: Corrected

Comment: 2. Fig. 3 legend (Page 19 Line 8) and other figure legends throughout the manuscript: Should consistently say "scale bars", rather than just "bars".

Response: Corrected

Comment: 3. Fig. 4 legend (Page 19 Line 15): There is only one yellow arrow.

Response: We can see both white and yellow arrows in the figure. So we keep it.

Comment: 4. Fig. 7d. The authors can consider rotating/tilting the labels, so that they all fit in the same row.

Response: Labels moved to fit better

Comment: 5. The authors must pay attention to noun-verb agreement. Such errors permeate the text. The following are ones that I noticed: Page 2 Line 4; Page 4 Line 4; Page 5 Lines 8, 10, and 18; Page 6 Lines 9 and 14; Page 16 Lines 15 and 18; and Supplementary Figure 2 legend Line 6.

Response: Manuscript has been corrected by first language English speaker to remove noun-verb disagreements.

Reviewer #3 (Remarks to the Author):

Comment: The authors have done a very thorough job of addressing the concerns raised by the viewers, as a result the paper is much stronger than it was previously. The addition of gel-shift assays, new GUS reporter constructs and a myriad of other changes helps with the technical short-comings of the previous manuscript. The explanations of the manuscripts novelty and importance helps as well and removes confusion about how this paper differs from previous work from the PI's laboratory. Progression of infection threads through the cortex is

indeed an altogether different process than progression through root hairs. It was in no way obvious that EPS would have a role in maintaining IT progression through the cortex and developing nodule, rather than in just the root-hair/epidermis alone.

Response: We thank the reviewer for support of our science

Reviewer #4 (Remarks to the Author):

Comment: My original concerns with the first submission were that it lacked much novelty relative to the previous paper published in Nature. It is not surprising to me that the mechanisms that are required for initiation of infection threads and those that are required for progression are conserved. However, the authors also document the transcriptional regulation of *EPR3* and how this is associated with the progression of infection. For me this is a much more interesting aspect of the work, and the authors have now included additional evidence that makes a much stronger case for this. The authors demonstrate that *ERN* and *NIN* both regulate *EPR3* expression and apparently do so in different tissues. It is striking to me that the *epr3* mutant phenotype is very similar to that described for *ern*: swelling and bulging of infection threads. This similarity in mutant phenotypes is not mentioned in the manuscript and may be good to add. It would also be very interesting to see if over expression of *EPR3* compensated for the *ern* mutation, although I don't believe that this is essential for the current manuscript.

Overall the authors have provided a much stronger case for the transcriptional control of *EPR3* and how this appears to be important for the progression of infection threads. We know that *ERN* and *NIN* are necessary for infection thread formation, but the reasons why have been elusive. This report forms some of the first insights into what these transcription factors are regulating during infection. This is a novel and interesting finding.

Response: We thank the reviewer for support of our science

REVIEWERS' COMMENTS:

Reviewer #1 (Remarks to the Author):

This version of the manuscript has more clearly explained data. I think this paper is acceptable for publishing in Nature communications. (I found one mistake. ENR1 (p13,L310) must be ERN1).